# A genome-wide association study reveals the relationship between human genetic variation and the nasal microbiome

Xiaomin Liu [1,2,9], Xin Tong[1,9], Leying Zou[1,9], Yanmei Ju[1,2,9], Mingliang Liu[1], Mo Han [1], Haorong Lu[3], Huanming Yang [1,4], Jian Wang[1,4], Yang Zong[1], Weibin Liu[1], Xun Xu [1], Xin Jin [1], Liang Xiao[1,5], Huijue Jia [6,7✉], Ruijin Guo [1✉] & Tao Zhang [8✉]

The nasal cavity harbors diverse microbiota that contributes to human health and respiratory diseases. However, whether and to what extent the host genome shapes the nasal microbiome remains largely unknown. Here, by dissecting the human genome and nasal metagenome data from 1401 healthy individuals, we demonstrated that the top three host genetic principal components strongly correlated with the nasal microbiota diversity and composition. The genetic association analyses identified 63 genome-wide significant loci affecting the nasal microbial taxa and functions, of which 2 loci reached study-wide significance ($p < 1.7 \times 10^{-10}$): rs73268759 within *CAMK2A* associated with genus *Actinomyces* and family Actinomycetaceae; and rs35211877 near *POM121L12* with *Gemella asaccharolytica*. In addition to respiratory-related diseases, the associated loci are mainly implicated in cardiometabolic or neuropsychiatric diseases. Functional analysis showed the associated genes were most significantly expressed in the nasal airway epithelium tissue and enriched in the calcium signaling and hippo signaling pathway. Further observational correlation and Mendelian randomization analyses consistently suggested the causal effects of *Serratia grimesii* and *Yokenella regensburgei* on cardiometabolic biomarkers (cystine, glutamic acid, and creatine). This study suggested that the host genome plays an important role in shaping the nasal microbiome.

[1] BGI Research, Shenzhen 518083, China. [2] College of Life Sciences, University of Chinese Academy of Sciences, Beijing 100049, China. [3] China National Genebank, BGI-Shenzhen, Shenzhen 518120, China. [4] James D. Watson Institute of Genome Sciences, Hangzhou 310058, China. [5] Shenzhen Engineering Laboratory of Detection and Intervention of Human Intestinal Microbiome, BGI-Shenzhen, Shenzhen 518083, China. [6] Greater Bay Area Institute of Precision Medicine, Guangzhou, Guangdong, China. [7] School of Life Sciences, Fudan University, Shanghai, China. [8] BGI Research, Wuhan 430074, China. [9] These authors contributed equally: Xiaomin Liu, Xin Tong, Leying Zou, and Yanmei Ju. ✉email: huijue_jia@fudan.edu.cn; guoruijin@genomics.cn; tao.zhang@genomics.cn

Bacteria in the nasal cavity have been primarily studied in the context of infection[1–9]. *Staphylococcus aureus* is a key pathogenic bacterium in chronic rhinosinusitis with nasal polyps[1], and its presence increases the risk of infection[2]. Reduced bacterial diversity in the nasal cavity has been linked to asthma, allergic rhinitis (AR), and asthma with AR comorbidity[3,4]. The nasal microbiota also affected host susceptibility to acute respiratory tract infections (ARTI)[5,6] and correlated with the clinical outcome of COVID-19[7]. *Moraxella catarrhalis*, *Haemophilus influenzae*, and *Streptococcus pneumoniae* have been shown to increase the risk of otitis media[8] and ARTI[9]. The nasal epithelium is being actively explored for drug delivery[10], especially given the olfactory defects seen for neurodegenerative diseases. In a murine model of influenza infection, intranasal administration of Bifidobacterium longum protected against lethal infections[11]. The outbreak of the SARS-CoV-2 pandemic has brought greater attention to understanding the significance of the respiratory tract microbiota. However, there have been fewer studies focusing on characterizing the microbiome profile of the upper respiratory tract, particularly the nostrils[12]. Therefore, it is now more important than ever to gain a comprehensive understanding of the nasal microbiome as an ecological community, including investigating the extent to which host genetics and other factors influence its composition.

Given the emerging evidence of human genetic influences on the faecal microbiome[13–19] and the oral microbiome[20,21], here we first aim to investigate whether human genetics also contribute to the nasal microbiome. Previous analysis using 16 S rRNA gene amplicon sequencing on twins did not find monozygotic twins to harbour a more similar nasal microbiota than dizygotic twins or unrelated pairs but reported twins concordance in the copy number per swab[22]. Another study focusing on 144 European adults identified very limited host genetic variations influencing upper airway microbial composition[23]. These two studies were investigated in individuals with a small sample size. Hence, the contribution of human genes to the composition and functions of the nasal microbiome remains largely unclear. Gaining insight into the factors including genetic and non-genetic that influence and characterize the nasal microbiome within a meticulously designed cohort is essential for comprehending both upper respiratory health and its broader implications for systemic well-being.

The 4D-SZ cohort is a carefully designed multi-omics cohort[19,20,24–27], comprising shotgun data of the metagenome from multiple body sites including the nasal cavity and host genome. It also incorporates information on metabolic traits, extensive questionnaires, and comprehensive clinical data. Based on the nasal metagenome and host genome data of 1401 healthy adults from the 4D-SZ cohort, we first estimated the impact of host genome on the nasal microbial community and demonstrated the great influence of host genetics principal components on the diversity and composition of the nasal microbiome. Then, we identified genome-wide and study-wide significant associations of host genetic loci with microbial taxa and functional pathways by performing metagenome-genome-wide association study (M-GWAS). We investigated the functional, tissue, and disease enrichments of the nasal microbiome-associated loci. Furthermore, using multi-omics data from the same 4D-SZ cohort, we compared the influence of host genetics and other host factors on shaping the nasal microbiome. Finally, we illustrated the impact of host metabolites such as cysteine on the nasal microbiome through the use of Mendelian randomization (MR).

## Results

### Host genetics correlated with the nasal microbiota diversity and composition. We set out by examining the correlation

between host genetic variations and the overall diversity of the nasal microbiome in 1401 individuals (63% females; a mean age of 30 years old) with high-depth metagenome data as well as integrated host genome data (a mean depth of 30× for host genome and an average sequencing data of 77.31 ± 23.00 Gb for nasal samples; Supplementary Data 1; Supplementary Fig. 1; Methods). The host genetic principal components (PCs) were examined, and the results revealed strong associations between the top three PCs and microbial α-diversity ($p < 0.05$ for both Shannon and Simpson indices based on species-level abundances, multivariable linear model; Fig. 1). Specifically, PC1 ($r = -0.07$, $p = 8.7 \times 10^{-3}$), PC2 ($r = -0.14$, $p = 1.39 \times 10^{-7}$), and PC3 ($r = -0.08$, $p = 1.8 \times 10^{-3}$) were markedly correlated with the Shannon index. This is consistent with previous analyses of low-depth human genome sequences from 93 individuals in the Human Microbiome Project (HMP), which reported PC1 to associate with α-diversity in the anterior nares[13]. We verify that the top two host genetic PCs in our cohort are strongly associated with self-reported ancestry, namely the geographical origin of their ancestry (commonly before and including grandfather) from northern or southern China ($p < 2.2 \times 10^{-16}$ for PC1 and $p = 1.78 \times 10^{-11}$ for PC2, Wilcoxon rank-sum test; Fig. 1b, c). We further confirmed that the individuals originating from northern China exhibited a higher nasal microbial α-diversity than those who originated from southern China (mean Shannon index of 1.35 vs 1.28; Wilcoxon Rank-Sum test; $p = 9.22 \times 10^{-3}$; Supplementary Fig. 2), despite the individuals themselves currently living in the same city. Also, the host genetic PC1 exhibited a correlation with the abundances of 14 nasal genera, including *Micrococcus*, *Anaerococcus*, *Elizabethkingia*, *Campylobacter*, *Finegoldia*, *Yokenella*, *Serratia*, *Streptococcus*, *Gemella*, *Staphylococcus*, *Actinomyces*, *Malassezia*, *Veillonella*, and *Prevotella* (Supplementary Data 2; Spearman test; False Discovery Rate (FDR) $p < 0.05$). Nine of these 14 PC1-associated genera also displayed differential abundances between the individuals deriving from China's northern and southern regions (Supplementary Data 2; $p < 0.05$).

Furthermore, the top three host genetic PCs showed correlations with four of the top ten microbiome β-diversity principal coordinates (PCos; computed based on species-level Bray–Curtis dissimilarity; $p < 0.05$ for pairwise associations, Spearman correlation; Fig. 1). Additionally, the top eight host genetic PCs (PC1-PC8) showed at least one correlation with any of the microbiome PCos among PCo2 to PCo8. These results suggested host genetics greatly influence nasal bacterial diversity and composition.

We also investigated associations between host genetic variants and nasal microbiome α-diversity and β-diversity (namely the top ten PCos). This analysis found two loci with marginal genome-wide significance ($p < 5 \times 10^{-8}$, Fig. 1e, Supplementary Data 3). The SNP rs77221359, located in the intronic region of *CACNB2*, was significantly associated with the Shannon index representing the α-diversity of nasal samples ($p = 2.6 \times 10^{-8}$; Supplementary Fig. 3). Notably, *CACNB2*, which encodes a subunit of a voltage-dependent calcium channel protein that is critical for mediating intracellular Ca2+ influx, has been established as a risk gene for psychiatric disorders such as schizophrenia and bipolar disorder[28,29]. Additionally, the SNP rs77221359 has been linked to multiple phenotypes or diseases, including chronic heart failure ($p = 0.007$), asthma ($p = 0.012$), and chronic sinusitis ($p = 0.015$), as reported in the biobank Japan (BBJ) database[30]. The second identified significant association was for SNP rs79409173, located in the intronic region of *WNK1*, with PCo6 ($p = 2.4 \times 10^{-8}$; Supplementary Data 3). The serine-threonine kinase *WNK1* functioning as a Cl⁻ sensor plays an important role in mature neuron development[31,32]. These are interesting associations, given the involvement of nasal microbiome in asthma, chronic sinusitis, and neurological diseases[33].

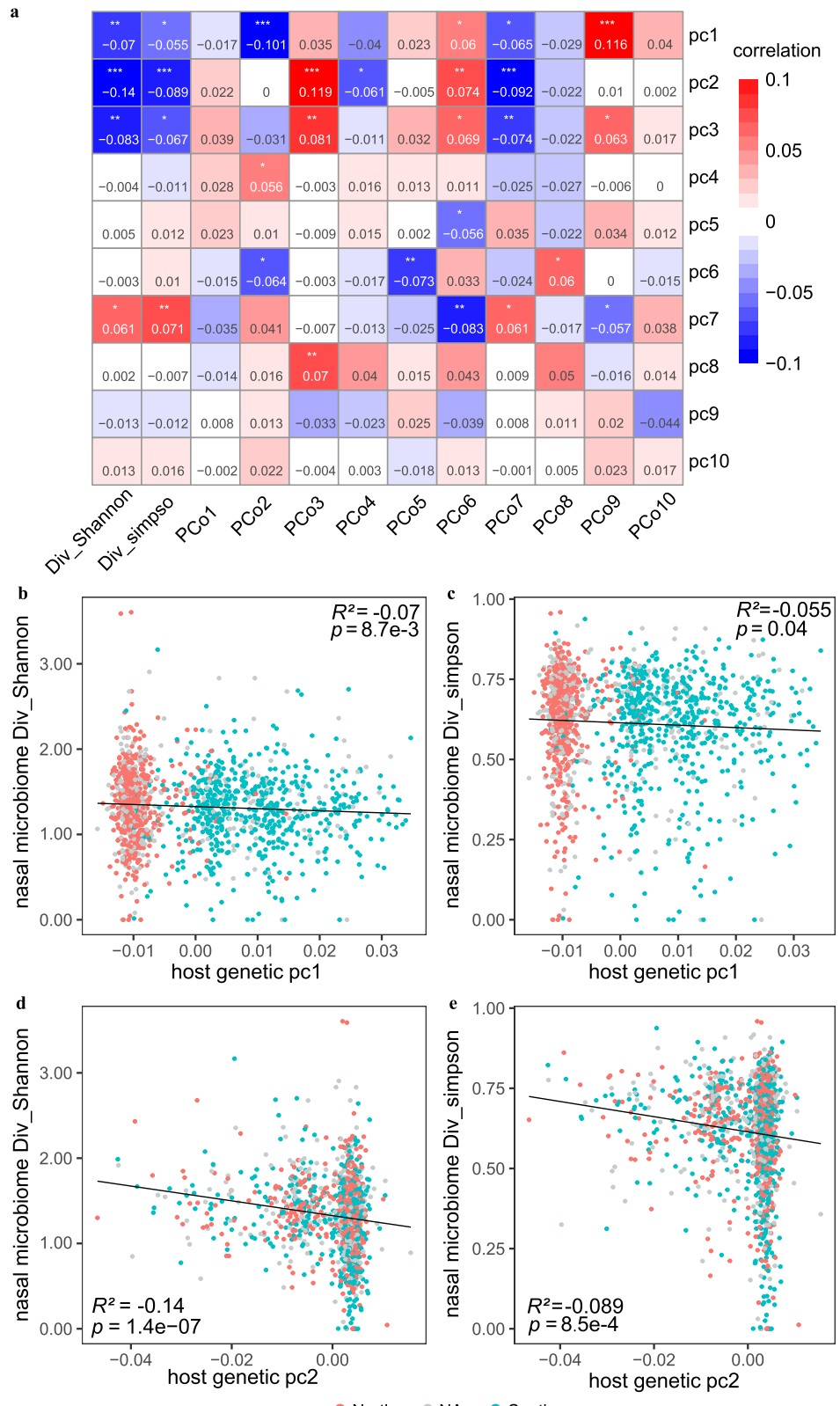

**Fig. 1 Host genetic principal components (PCs) associated with the nasal microbial diversity and compositions. a** Correlation ($R^2$) estimates of ten host PCs (PC1-PC10) with the nasal microbial diversity (Shannon and Simpson indices) and compositions (PCo1-PCo10), evaluated using a linear model. Red colours denoted the positive correlation and blue denotes the negative correlation. *$p < 0.05$, **$p < 0.01$ and ***$p < 0.001$. **b**, **c** indicate the correlations between the host genetic PC1 and the nasal microbial Shannon index (**b**) and Simpson index (**c**), respectively. **d**, **e** indicate the correlations between the host genetic PC2 and the nasal microbial Shannon index (**d**) and Simpson index (**e**), respectively. The individuals whose ancestry lived in the northern or the southern China were marked in red and blue, respectively. Grey dots represented individuals with the unknown ("NA") ancestry information.

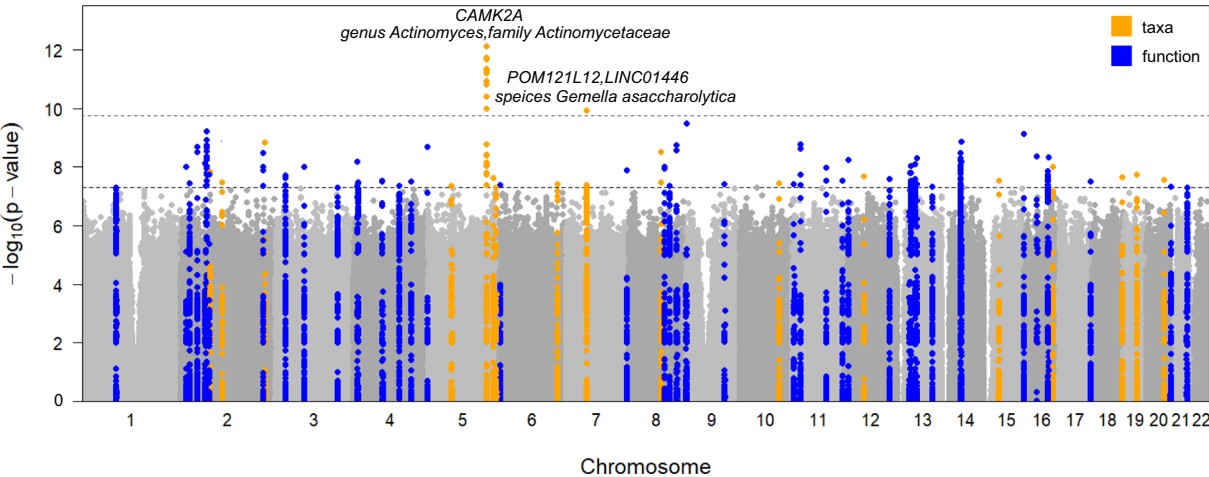

**Fig. 2 Host genetic signals associated with nasal microbial taxa and functions.** Manhattan plot shows the genetic variants associated with the nasal microbial taxa ($n = 86$) and functions ($n = 207$). The horizontal grey and black lines represent the genome-wide ($p = 5 \times 10^{-8}$) and study-wide ($p = 1.71 \times 10^{-10}$ for 293 independent M-GWAS tests) significance levels, respectively. Seventeen loci associated with taxa and 46 loci associated with functions reaching genome-wide significance were marked in orange and blue, respectively. The top two loci, their located genes, and associated nasal taxa that reached study-wide significance were also listed.

**Host genetics strongly associated with nasal bacterial or fungal taxa and functions.** With this large cohort of both whole genome and whole metagenome data, we next conducted M-GWAS on the nasal microbiome. The microbial taxonomy and function profiles were determined by aligning metagenomic reads to marker genes according to metaphlan3 and humann3, respectively (Supplementary Fig. 3a-b). After filtering highly correlated features, we obtained 293 independent nasal microbial features (Supplementary Data 4; 86 taxa and 207 functions; $r^2 < 0.995$ using a greedy algorithm, **Methods**). The M-GWAS analyses were performed on 7 million human genetic variants (minor allele frequency (MAF) ≥ 1%) by adjusting for age, gender, body mass index (BMI), sequencing read counts, and the top ten host PCs. Our GWAS analyses did not demonstrate any evidence of an excess false positive rate (genomic inflation factors $\lambda_{GC}$ ranged from 0.979 to 1.054, with a median of 1.012; Supplementary Fig. 4).

In total, we identified 180 independent associations involving 63 independent loci (distance < 1 Mb and $r^2 < 0.2$) and 46 independent taxa/functions reaching genome-wide significance ($p < 5 \times 10^{-8}$; Fig. 2 and Supplementary Data 5). Out of the 63 loci we identified, 17 were associated with 14 microbial taxa and 47 were associated with 32 bacterial functions. Specifically, seven of the 63 loci were associated with at least two independent taxa or functions (Fig. 2 and Supplementary Data 5). All genome-wide significant associations were listed in Supplementary Data 5. The 63 genome-wide loci explained a higher average fraction of variance ($R^2$) for microbial functions (mean $R^2$ of 8.7%; ranging from 2.49% for PWY-5138: unsaturated, even numbered fatty acid β-oxidation to 19.01% for ANAGLYCOLYSIS-PWY: glycolysis III (from glucose); Supplementary Data 6) compared to microbial taxa (mean $R^2$ of 5.6%; ranging from 2.35% for species *Malassezia globosa* to 12.48% for class Actinobacteria).

With a more conservative *Bonferroni*-corrected study-wide significant $p$-value of $1.71 \times 10^{-10}$ ($= 5 \times 10^{-8}/293$ for 293 M-GWAS tests), we identified 2 genomic loci associated with 3 nasal microbial taxa (Fig. 2).

The strongest association was observed for SNP rs73268759 located in the intronic region of the *CAMK2A* gene, with minor allele C (MAF = 0.021) positively associated with the presence/absence phenotype of genus *Actinomyces* (Fig. 3a-b; $p = 7.75 \times 10^{-13}$) and family Actinomycetaceae ($p = 2.06 \times 10^{-12}$;

spearman $r = 0.970$ with *Actinomyces*). When further testing for the association of rs73268759 with the relative abundance of *Actinomyces* presented in 162 individuals, this association was more significant (Fig. 3c; $p = 9.69 \times 10^{-15}$). The association was also replicated in the gut, with rs73268759 associated with the relative abundances of the gut-derived *Actinomyces* (Fig. 3d; $p = 0.014$). *CAMK2A* encodes a protein belonging to the Calcium/calmodulin-dependent protein kinase II (CAMKII), and its oxidation promotes asthma through the activation of mast cells[34]. The *CAMK2A*-associated two taxa, Actinomycetaceae and its main genus *Actinomyces*, are abundant commensals of the human oropharynx and have been increasingly associated with infections at many body sites[35]. We found these two taxa were also positively correlated with an increased risk of oral diseases (e.g., oral ulcers and caries), upper respiratory tract infection and urinary system infection (Fig. 3f; multivariable linear regression $p < 0.05$), when checking the correlation between the two taxa and phenotypic traits in this cohort. Collectively, these results supported a link between the genetic variation in the *CAMK2A* gene, abundances of the two taxa, and the inflammatory response to the upper respiratory tract.

The second strongest association was seen on rs35211877, which is a deletion of allele T (MAF = 0.057) located 163 kb downstream of the *POM121L12* gene. This deletion was negatively associated with *Gemella asaccharolytica* (Fig. 2; $p = 1.13 \times 10^{-10}$). Furthermore, this deletion was also found to be associated with increased risks of asthma ($p = 0.005$), epilepsy ($p = 0.005$), and gastroesophageal reflux disease ($p = 0.012$) when searching the BBJ database. We also found that *Gemella asaccharolytica* had a positive correlation with stress sources ($p = 1.07 \times 10^{-4}$), frequently allergic sub-health status ($p = 0.003$), and gastritis ($p = 0.023$), but a negative correlation with vitamin D and D2 levels in this cohort (Supplementary Fig. 5).

*Cutibacterium* was the most abundant genus in the nose (Supplementary Fig. 6), and it was linked to the SNP rs186899741 located in the intergenic region of *OCSTAMP* and *SLC13A3* ($p = 2.62 \times 10^{-8}$). This SNP was associated with 25-hydroxyvitamin-D3 ($p = 0.001$) in this cohort and type 2 diabetes ($p = 0.006$) in the BBJ. The second most abundant genus *Corynebacterium* (Supplementary Fig. 6) was linked to the SNP rs117538984 located in the intronic region of *BARD1* ($p = 1.44 \times 10^{-9}$), which was associated with colorectal cancer

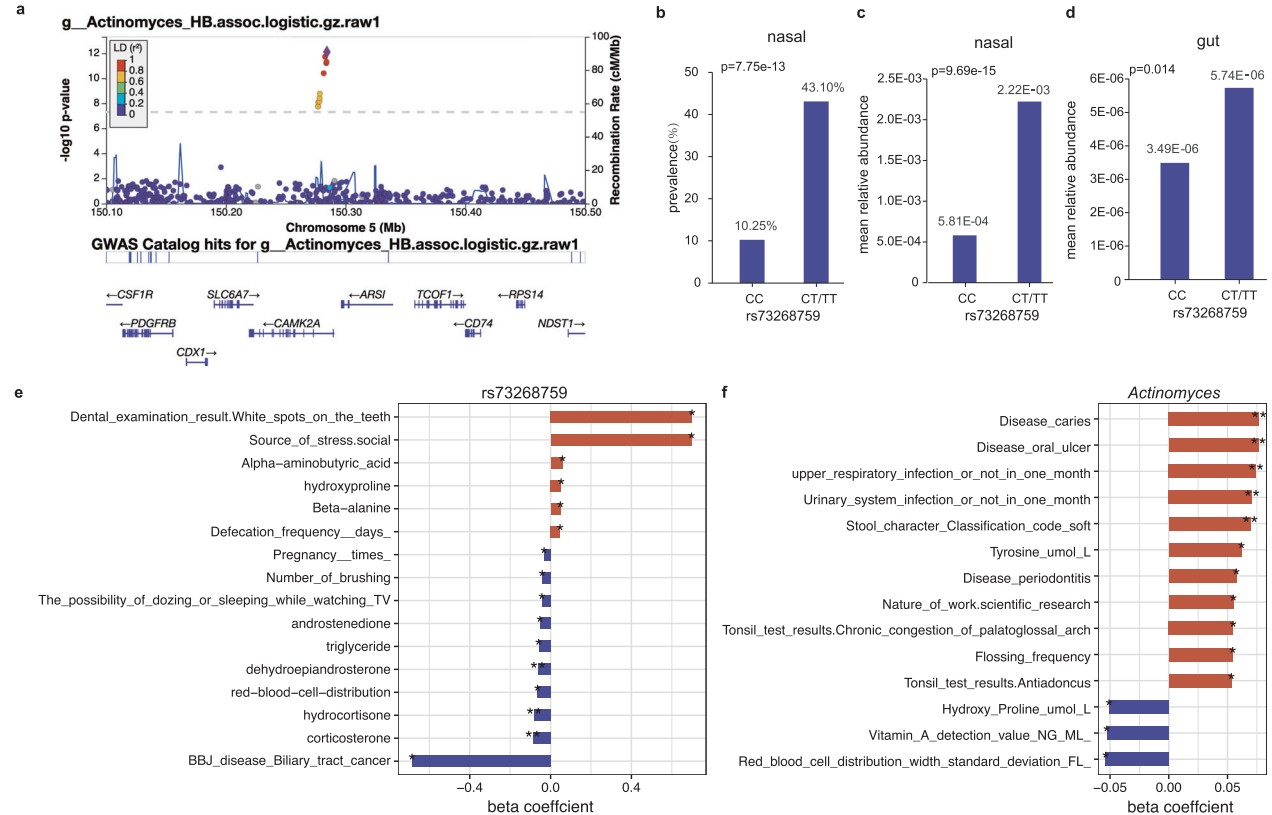

**Fig. 3 The links among human *CAMK2A* locus (rs73268759), the nasal genus *Actinomyces*, and host traits in this cohort. a** The regional plot presented the strongest association of *CAMK2A* variation with the genus *Actinomyces* (presence/absence status; $p = 7.75 \times 10^{-13}$). **b** The prevalences of the genus *Actinomyces* were compared between different genotyped individuals (CC vs CT/TT) according to *CAMK2A* (rs73268759) variation. The *p*-value in **b** was obtained using a logistic model based on the presence/absence status of the genus *Actinomyces*. **c, d** The mean relative abundances of the genus *Actinomyces* from the nasal (**c**) and gut (**d**) sites, compared between different genotyped individuals (CC vs CT/TT) according to *CAMK2A* (rs73268759) variation. The statistical comparisons in (**c**) and (**d**) denote the *p*-values of the linear regression analysis on log-transformed relative abundances. All analyses were performed by adjusting sex, age, BMI, sequencing reads, and the top ten PCs. **e** The bar plot showed associations of *the CAMK2A* (rs73268759) variation with host traits in this cohort. **f** The correlations of the genus *Actinomyces* with host traits in this cohort. In **e**, **f**, only associations or correlations with *p* < 0.05 were shown. Significant code: 0.05 * 0.01 ** 0.001. The source data for e-f was available in Supplementary Data 15.

($p = 0.002$) and pulmonary tuberculosis ($p = 0.01$) in the BBJ. Interestingly, *BARD1* is also known to interact with *BRCA1*[36], and has been reported to be elevated in the urine of breast cancer patients compared to controls[37]. In addition, we found several associations involving bacteria commonly detected in other body sites, for example, rs139288082 in *UST* with *Streptococcus oralis* (an oral commensal, belongs to the mitis group of streptococci and occasionally causes opportunistic infections such as bacteremia and endocarditis[38]), rs2716569 in the intergenic region of *LINC01568 - LOC101928035* with *Malassezia globosa* (a traditionally known fungus for the skin microbiome, implicated in conditions such as inflammatory bowel diseases[39], lung infections, and breast cancer[40]).

In addition to the associations with nasal microbial taxa, 46 host genetic loci were linked to nasal functions. For example, ANAGLYCOLYSIS-PWY: glycolysis III (from glucose) was linked to three gene loci, including *PUM3*, *NELL1*, and *LINC02580*. Both *PUM3*[41] and *NELL1*[42] exhibited the ability to regulate cell proliferation. COLANSYN-PWY: colonic acid building blocks biosynthesis was linked to multiple SNPs in the *MEIS1* gene ($p = 5.70 \times 10^{-10}$; associated with monocyte count[43]). PWY-5686: UMP biosynthesis was linked to *LMF1* ($p = 7.36 \times 10^{-10}$), which harbours variants associated with extreme respiratory outcomes following preterm birth[44]. Compared with the taxa associations, the interpretation of associated functional pathways as a single factor pointing to a specific

biological mechanism is challenging due to their complexity. These associations and their underlying interaction mechanisms called for further verification and investigation in future studies.

**PheWAS and gene expression analysis for 63 microbiome-associated loci**. To better understand the potential biological mechanism of the 63 nasal microbiome-associated variants (MAVs), we first explore their associations with diseases and traits. The phenome-wide association study (PheWAS) conducted on this cohort and the BBJ revealed that these 63 MAVs were enriched in 24 host-related traits and diseases: five diseases (asthma, type 2 diabetes(T2D), colorectal cancer, atopic dermatitis, abortion), glucose and low-density lipoprotein (LDL) from the BBJ cohort; six metabolic-related traits including chromium, phenylalanine, triglyceride, vitamin A, lymphocytes count and diastolic pressure; as well as health conditions (sleep quality, sub-health status) and lifestyles in this cohort (fisher exact test $p < 0.05$; Fig. 4).

As the 63 MAVs are mostly located in the intronic or intergenic region, we annotated their associations with gene expression in a recently published nasal airway epithelium transcriptome data[45] and across the 49 tissues in the Genotype-Tissue Expression (GTEx) database[46]. 26 of the 63 MAVs were mapped to 33 genes (intronic or <5 KB upstream/downstream). We investigated the expressions of the 33 top genes across 50 tissues and found that over half (>16) of the genes significantly

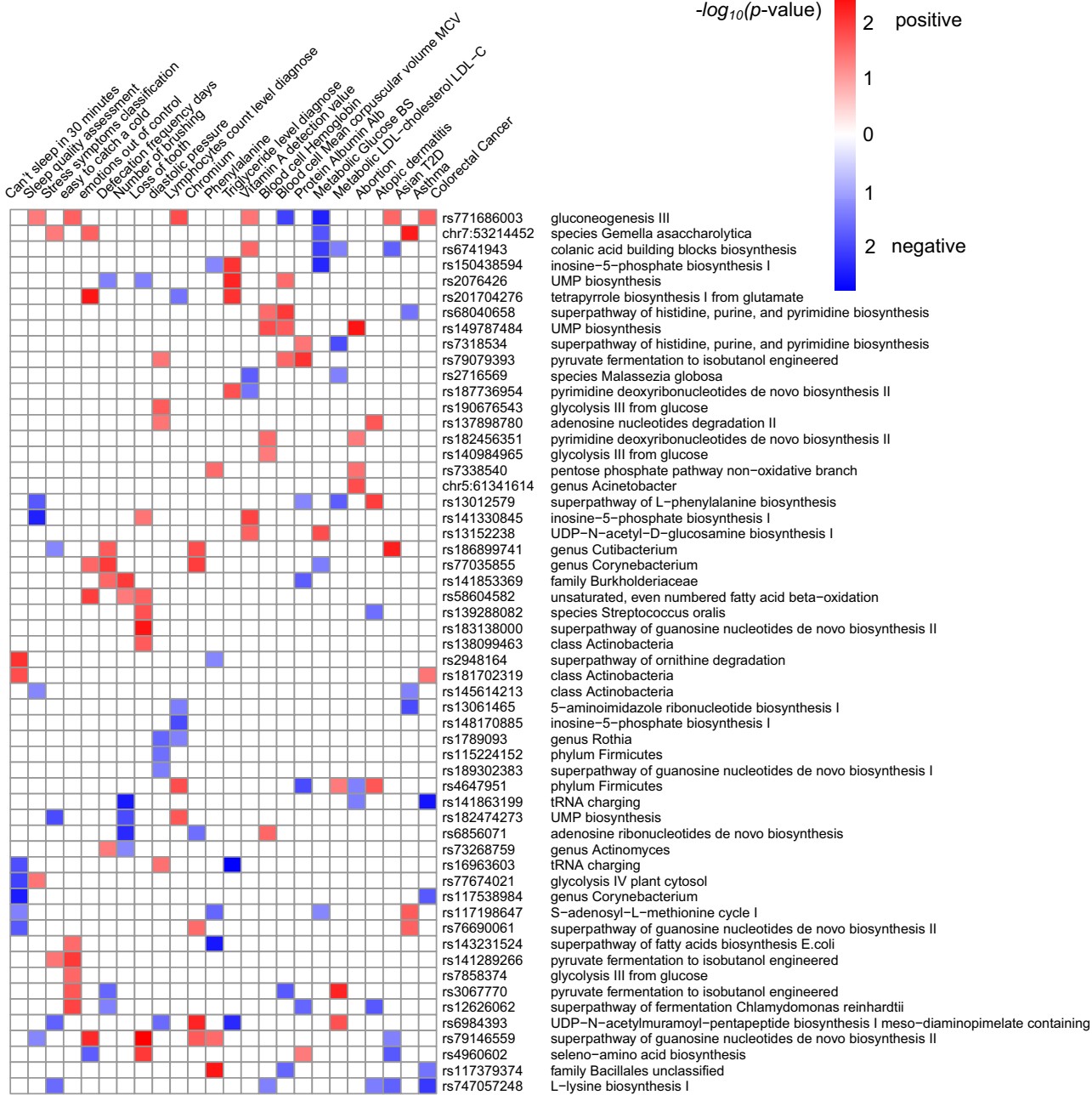

**Fig. 4 A phenome-wide association study presented the enrichment results of top MAVs in traits or diseases.** The heatmap plot showed associations of index SNPs of the top 63 loci with traits in this cohort or diseases in the BBJ cohort. 56 of the top 63 SNPs showed enrichments and were listed, and only traits or diseases with significantly enriched $p < 0.05$ are shown. The enriched $p$-value was calculated using the fisher exacted test. The red indicted the positive association and the blue indicted the negative association between SNP and corresponding trait.

demonstrated expression in seven of the 50 tissues (Fig. 5a). As expected, the nasal airway epithelium exhibited the strongest enrichment, with 23 of the 33 genes showing significantly expressed ($p < 10^{-4}$; Supplementary Data 7). The tissues such as thyroid, muscle skeletal, lung, testis, nerve tibial, and oesophagus muscularis, which are also enriched tissues shared by the gut MAV eQTL target genes[47], also showed enrichment. *BARD1* (associated with the abundance of genus *Corynebacterium*) and *LMF1* (associated with the abundance of PWY-5686: UMP biosynthesis) were the top two most expressed genes because of cumulative representation across 50 tissues (Supplementary Fig. 7). The strongest M-GWAS signal gene *CAMK2A* is enriched in 17 tissues. The hierarchical clustering showed several MAV top

genes (*ALMS1*, *ESD*, *ARHGAP10*, *MBP*) were more significantly expressed in the nasal airway epithelium than the other tissues (Fig. 5b).

We further expanded the genetic associations with suggestive $p < 10^{-6}$ (Supplementary Data 8) and performed gene expression analysis in the tissues, followed by gene functional mapping and disease enrichment analysis with the FUMA[48] platform (**Methods**). Those nasal MAVs of suggestive significance were mapped to 413 genes (<5 Kb for associated genetic loci). These genes exhibited enrichment for differentially expressed in multiple tissues including the most significant enrichment in the nasal airway epithelium and thyroid (Supplementary Fig. 8), in agreement with the findings using the genome-wide significant MAVs.

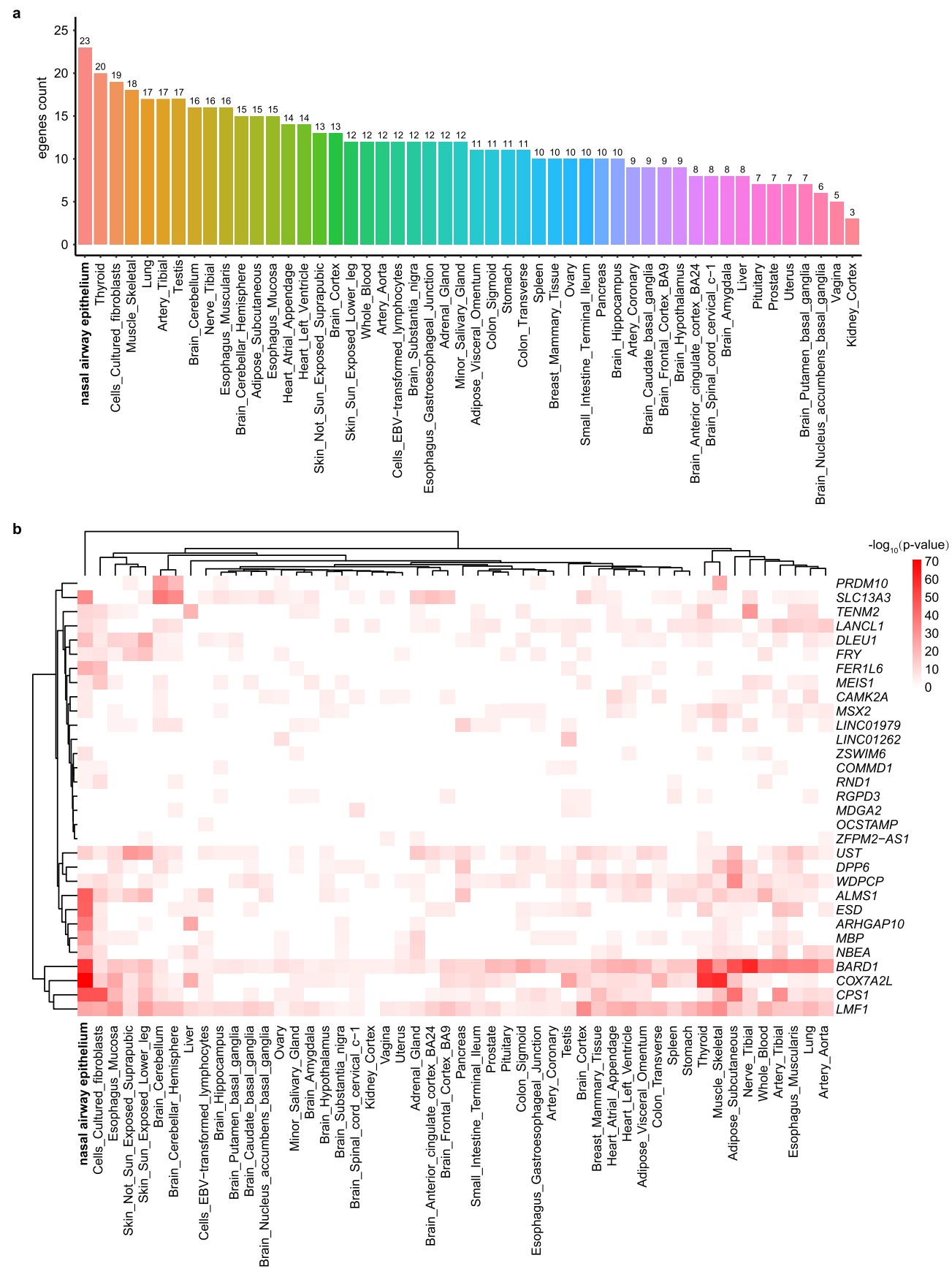

**Fig. 5 Nasal MAVs were enriched in the nasal airway epithelium and other relevant tissues. a**. The barplot showed the enrichment of 33 MAV genes (intronic or <5KB upstream/downstream) across the 50 tissue groups. **b**. The heatmap displayed the gene expression values of 33 MAV genes across the 50 tissue groups. The expression values come from a nasal airway epithelium transcriptome dataset and the GTEx v8 datasets. Cells filled in red represent higher expression compared to cells filled in white across genes and tissues.

Gene functional mapping with FUMA found that the gene sets of suggestive MAVs were mainly enriched in calcium signaling and regulating hippo signaling pathways, by using the KEGG, Wikipathways, and GO databases (Supplementary Fig. 9). The disease-enriched analysis in the GWAS catalogue using FUMA showed that the MAVs were enriched in 75 traits ($p_{\text{bon}} < 0.01$; Supplementary Fig. 10), with the strongest link to cardiometabolic related traits such as obesity-related traits, systolic blood pressure, diastolic blood pressure, body mass index, platelet count. It was also found that MAVs were involved in neuropsychiatric symptoms such as cognitive decline rate in late mild cognitive impairment, general cognitive ability, schizophrenia, dementia and core Alzheimer's disease neuropathologic changes (Supplementary Fig. 10). Correlations between MAVs with asthma, T2D, and colorectal cancer were confirmed through both the PheWAS analysis of BBJ and the bigger GWAS dataset using FUMA. Links between MAVs and diastolic pressure and sleep phenotypes were also replicated in this study and that of in GWAS catalogue studies using FUMA tool. Interestingly, MAVs showed correlations with smoking phenotypes (smoking status and smoking initiation), which have been reported to be key determinants of the airway microbiome[49,50]. This result suggests individual genes may influence the nasal microbes by making an individual genetically more likely to smoke or not, which needs to further validate in other cohorts due to the rarity of smokers in this current cohort.

**Host genetics and other factors contributed comparably to the nasal microbiome.** The above-identified 63 genome-wide significant loci could infer 9.51% of the variance in the nasal microbiome β-diversity. Applying association analysis for host genetic variants and microbiome β-diversity, we identified 21 loci with suggestive significance ($p < 10^{-5}$; Supplementary Data 9). Among the top index variants, eight were associated with *Staphylococcus epidermidis* ($n = 4$) or *Corynebacterium accolens* ($n = 4$), which were among the most abundant commensal in the nose (Supplementary Data 4) and might possess antimicrobial activity against pathogens[51,52]. The 21 top index variants that were most closely associated with β-diversity, while not reaching genome-wide significance, could explain 10.59% of the variance in the community structure, with each of them contributing from 0.34% to 0.70% of the variation (Supplementary Fig. 11). This observed $R^2$ (10.5%) under the real data was significantly greater than the average $R^2$ (7.4%) of 100 permutations ($P < 2.2 \times 10^{-26}$, t-test, this p represented the fraction of permutations in which the fraction of inferred variance was greater than observed). This fraction was lower than the proportion of host genetics influencing the gut microbiome (~20%) but close to that of the oral microbiome (10.14% ~ 14.14%), as previously inferred in the same cohort[19,20]. This is expected as the nasal cavity is more open compared to the half-closed oral and fully closed gut environments.

After establishing the contribution of host genetics to nasal microbiome composition and functions, we investigated the extent to which environmental factors influence the nasal microbiome compared to genetics. Among the 340 host traits (age, gender, BMI, dietary, lifestyle, health status questions, and blood measurements), 45 were significantly associated with β-diversity (BH-adjusted FDR < 0.05), via PERMANOVA

analysis (Supplementary Data 10). The five strongest associated factors observed were sex, serum testosterone and estradiol, serum iron concentration, and muscle mass, each explaining a variance of 0.65% to 1.47%. In total, the 45 significant host factors could infer 10.76% of the variance in the nasal microbiome β-diversity. This analysis showed that the contribution of host genetics to the nasal microbiome may be comparable to the other host factors. Host genetics together with other host factors collectively explained 19.19% of the variance in the nasal microbiome community.

**From observational correlation to Mendelian randomization for the nasal microbiome and host traits.** Host factors greatly influenced the nasal microbiome composition, we further investigated the correlation of trait-microbiota pairs for the 293 unique nasal microbial features and 340 host traits using multivariate linear regression. After adjustment for gender, age, BMI, depth, and the top four PCs, we observed 402 significant associations (Benjamini–Hochberg (BH)-adjusted $P < 0.05$, Supplementary Data 11). The genus *Elizabethkingia* and its species *E. miricola*, *E. bruuniana*, as well as genus *Serratia* and its species *S. grimesii*, showed the most links to the host traits ($n = 40, 34, 33, 27$ and 30, respectively, Supplementary Fig. 12). In return, creatine, cystine, aspartic acid, chromium, magnesium, and cystathionine were among the metabolites associated with the largest number of microbial features ($n = 42, 33, 22, 20, 20$ and 17, respectively, Supplementary Fig. 12).

Since top associated genetics variants showed strong power as instrumental variables (Supplementary Fig. 13) and well explained for both the nasal microbiome and host traits (Supplementary Fig. 14, Methods), as well as the strong correlations between them, we next performed bidirectional one-sample Mendelian randomization analysis for the 402 observationally significant associations, aiming to reveal the potential causal relationships between the nasal microbial features and host traits. In total, we identified 128 suggestive causal effects with $p < 0.05$, of which 4 were significant after Bonferroni correction ($p < 1.24 \times 10^{-4} = 0.05/402$; Fig. 6, Supplementary Data 12). The strongest MR evidence lies in *Serratia grimesii* that was causally associated with an increased cystine level ($\beta = 0.42$, $p = 1.34 \times 10^{-5}$). Cystine is formed from the jointing of two cysteine molecules, and cysteine metabolism genes have been reported to be widely present in the *S. grimesii* BXF1[53]. Moreover, the growth of *Serratia grimesii* could be well predicted under a given cysteine environment in simulations that used the gapseq model[54] (Supplementary Fig. 15). Compared to *Serratia grimesii*, *Streptococcus oralis* that showed no correlation with the cystine level in the observational and MR analysis was not influenced by adding the cystine in the simulation (Supplementary Fig. 15). In addition, *Yokenella regensburgei* was inferred to causally reduced glutamic acid and creatine concentrations. Further investigation of the genomics functional modules[55] confirmed *Serratia grimesii* was widely involved in cysteine and *Yokenella regensburgei* was widely involved in glutamic acid (glutamate) metabolic related pathways (Supplementary Fig. 16), supporting the MR inferences. The microbial superpathway of guanosine nucleotides de novo biosynthesis II was negatively associated with the dental condition of loss of tooth.

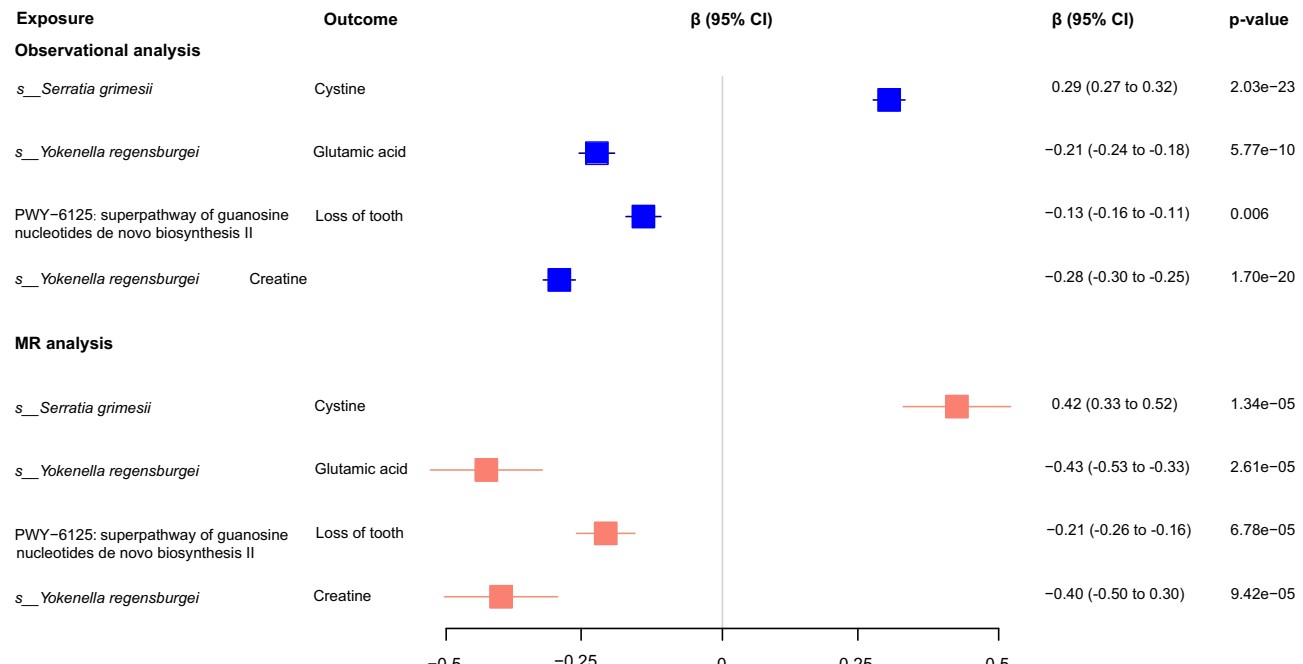

**Fig. 6 Mendelian randomization analysis identifying 4 causal relationships for the nasal microbial features and host traits (mainly metabolites) that were significant after *Bonferroni* correction.** Forest plot (in blue) showed the observational phenotypic correlations between the nasal microbial features and host traits (mainly metabolites), as calculated using the multivariate linear analysis corresponding to Supplementary Data 11. The β coefficient (95% CI) and BH adjusted *p*-value were shown. Forest plot (in red) represented the causal effects of microbial features on host traits (mainly metabolites), as calculated using the one-sample MR analysis corresponding to Supplementary Data 12. The β coefficient (95% CI) and *p*-value were shown.

## Discussion

In this study, we performed a large-scale M-GWAS for the nasal microbiome and reported abundant human genetic determinants for the nasal microbial taxa, functions, diversities, and compositions. The M-GWAS analyses identified a total of 63 genome-wide and 2 study-wide significant signals associated with the nasal microbiome, highlighting the power of high-depth metagenomics data and integrated high-quality host whole genome data in such a relatively larger nasal sample ($N = 1401$) compared to previous related studies[22,23]. These associations invited more and larger high-depth genome cohorts for further confirmation in the future.

Our study observed a high correlation between host PCs and nasal microbial diversity. This is an interesting and yet puzzling result because it could be really pointing towards evidence of co-evolution of the microbiome and their host[56,57], but it could also be yet another proof of unobserved correlation structures specific to the population. Although one paper indicated possible bias in the PCA analysis[58], it is important to note that the PCA analysis in this study depicted the Chinese North-South genetic structure (Fig. 1b,c) and showed Chinese obviously separated from the AFR and EUR populations (Supplementary Fig. 18). In addition to a great contribution to the nasal microbial diversity, host genetics was demonstrated to have impacts on the microbial composition, which extends a previous study that only reported host genetic influence on nasal bacterial diversity[22]. Through our analysis, we identified two microbiome-associated genes involved in the calcium signaling pathway: *CACNB2*, which is associated with nasal microbial alpha diversity, and *CAMK2B*, which is associated with the nasal bacteria *Actinomyces*. *CACNB2* has been implicated in psychiatric disorders[28,29]. The CamKIIα MD thalamic neurons (MD$^{CamKIIα}$) could act and cause arousal in mice from slow-wave sleep[59]. These findings suggesting the potential role of nasal microbiota in neuropsychiatric diseases. Furthermore, our PheWAS analysis revealed a link between MAVs and

neuropsychiatric traits, further supporting the potential impact of nasal microbiota in these disorders including the therapeutic effect and pathogenicity[60]. Regretfully, due to limited access to data on neuropsychiatric diseases for the Asian population, we were only able to observe a suggestive causal effect of Firmicutes on Schizophrenia through MR analysis (beta = 0.145, $p = 0.25$). We also confirmed the nasal microbiome-associated genes showed the strongest expression enrichment in the nasal airway epithelium and secondly in the thyroid. The connection between the nasal cavity and the thyroid has been well-known for a century[61,62].

Although the cohort is not yet very large for GWAS, the absence and presence of some associations are nonetheless notable for the nasal community. We identified a limited influence of host genetics on *Staphylococcus spp.*, in line with previous studies which reported host genetic factors exhibited only a modest influence on the *S. aureus* nasal colonization[63,64]. Lactobacilli, including those that are commonly attributed to the vagina, are prominently detectable in this and other respiratory studies[65,66]. We, however, have not seen a significant genetic association for Lactobacilli (Fig. 2 and Supplementary Data 4, 5). *Dolosigranulum pigrum*, a common and candidate beneficial nasal bacterium that inhibited *S. aureus* in vitro[66–68], also showed no significant genetic associations. In contrast, the other predominant and benign nasal bacterium *Corynebacterium* that inhibited the growth of *S. pneumoniae*[69,70] exhibited a strong host genetic attribute with genes *BARD1* and *TENM2*. *Acinetobacter*, whose abundance in the skin was influenced by host functions[71], presented a positive correlation with gene *ZSWIM6* in this nasal M-GWAS analysis. Likewise, *Micrococcus*, a taxon found in the foetal intestine and the upper reproductive tract[72–74], also showed genetic associations in niches of the nasal cavity and skin[71]. *Corynebacterium*, *Acinetobacter* and *Micrococcus* were recently proposed to be inherited in the mother-to-infants microbiota transmission[75].

Due to the continued success of dietary and faecal transplant studies in gnotobiotic mice, the gut microbiome is long believed to be a dynamic environmental factor. Our study, along with that of others[13–15,17,19,24,76–80], has identified genetic factors for gut microbiome taxa and functions. Besides the well-known story of lactose (*LCT*) and Bifidobacterium, taxa underlying the controversial concept of enterotypes, *Prevotella* and *Bacteroides*, also show human genetic associations[15,19]. A previous study identified 42 SNPs that together explained 10% of the variance of gut microbiome β-diversity[17]. Our prior studies based on the 4D-SZ cohort indicated host genetics explained great variances in the gut ($R^2 = 20.6\%$)[19] and oral microbiome ($R^2 = 14.14$ and 10.14% for salivary and tongue dorsum microbiome, respectively)[20]. With these high-depth nasal metagenomic sequencing samples and integrated high-quality host whole genome data also from the same 4D-SZ cohort, we found that as few as 21 genetic loci (10.59%) could explain the nearly same fraction of the β-diversity variance for the nasal microbiome compared to the 45 significant host factors (10.76%). These results consistently confirmed the important role of host genes in shaping the human microbiota. The nasal cavity serves as an entry point and a reservoir for pathogens, making it an important site to consider when addressing respiratory infections. By understanding the specific genetic and host factors that influence the nasal microbiome composition, we can potentially design interventions that target and modify these factors to promote a healthier microbiome. This personalized approach may lead to improved outcomes and more tailored treatments for individuals. Moreover, by considering the interplay between host genetics and the nasal microbiome, we can gain insights into the mechanisms underlying respiratory diseases and their potential treatments. This knowledge may also help identify individuals who are more susceptible to certain conditions, allowing for early interventions or preventive measures.

As we were fortunate to have all the data in the same cohort, we first examined the phenotypic correlations between microbes and metabolites and then determined the directionality of these correlations. Observational correlation could be treated as a prerequisite for strong causality. Observational and MR analyses consistently identified 4 significant causal relationships after Bonferroni correction. Specifically, we found that two species, *Serratia grimesii* and *Yokenella regensburgei*, were causally associated with three cardiometabolic-related biomarkers. *Serratia grimesii* influenced cystine levels, while *Yokenella regensburgei* influenced glutamic acid and creatine concentrations. Cystine is the most heritable amino acid in this Chinese cohort ($h^2 = 0.41$)[24], and it is likely important for the redox environment and the vasodilating effect of released H2S. Glutamic acid has been reported as a central currency in host and microbiome metabolism[24,25]. These findings shed light on the causal relationships between specific microbes and metabolites related to cardiometabolic health.

This study has several potential limitations. First, although our study represents the first large nasal M-GWAS conducted thus far, the sample size remains relatively small, similar to early initiated gut M-GWAS studies[13,15,18]. Unlike host GWAS and MR studies that benefit from cohorts (UKB[81], BBJ[82], etc.) with hundreds of thousands of participants, limited sample size of microbiome GWAS reduces statistical power and increases the likelihood of spurious associations. This is an issue in all current M-GWAS studies and could be improved as more microbiome data and host genome increased. Second, in addition to the two study-wide significant associations found in this M-GWAS studies, we have also reported associations that reach the genome-wide significance level. Nevertheless, the lack of simultaneous nasal microbiome and host whole genome data impedes the replication of most GWAS results and severely hinders the interpretation of MR results. Furthermore, it is important to note that the results presented in this study, similar to most previous gut microbiome studies[80,83], lack a clear understanding of the underlying mechanisms through which the microbiome may causally influence the outcomes. Therefore, further replications of results in independent cohorts and explorations of relevant mechanisms are required in the future. Finally, the use of principal components (PCs) and principal coordinate analysis (PCoA) as markers for population structure and microbiome structure may introduce biases. While these approaches offer valuable insights into the genetic and microbial variations among individuals, they may not capture the full complexity of these structures. Additionally, we integrated blood-derived host genome reads and nasal samples-extracted host genome reads to gather the individual's host genome data. While the average genotype concordance between the direct blood WGS data and integrated WGS data was 98.15%, indicating minimal bias when using the integrated WGS data as a substitute for blood WGS data, inherent biases may still exist. However, the high genotype concordance and absence of population stratification shown in the PCA analysis demonstrated the feasibility of reconstructing personal genome information from human genome reads in metagenome sequencing data[84].

In summary, we demonstrated that host genetic attributes play an important role in shaping the nasal microbiome, not only the gut and oral microbiome. The identified abundant causal relationships between the nasal microbiome and host metabolites suggested the potential clinical application of targeting the nasal microbiome. The applications with the nasal microbiome are perhaps more exciting given the link between olfaction and brain development.

## Methods

**Study subjects**. All the individuals in this study were part of the '4D-SZ' cohort, with blood, oral and nasal samples and extensive metadata collected for a multi-omics study as previously reported[19,20,24–27,66]. In this study, 1,593 nasal samples from the cohort were collected for whole metagenomic sequencing in 2018 (Supplementary Data 1). 1,457 of the 1,593 individuals also had blood samples with whole genome sequencing. The protocols for blood and nasal collection, as well as the whole genome and metagenomic sequencing, were similar to our previous literature[19,25,26,66]. For the blood sample, DNA was extracted using MagPure Buffy Coat DNA Midi KF Kit (no. D3537-02) according to the manufacturer's protocol. For each nasal sample collected by the anterior nare swab, a 2 ml stabilizing reagent kit was used and DNA was extracted using MagPure Stool DNA KF Kit B (no. MD5115-02B). The protocol includes a step of mechanical cell disruption by bead beating and optimizes the extraction process of DNA from both bacterial and fungal cells[85]. The DNA concentrations from blood and nasal samples were estimated by Qubit (Invitrogen). 500 ng of input DNA from blood and nasal samples were used for library preparation and then processed for paired-end 150 bp sequencing using the DNBSEQ platform[86].

All recruitment and study procedures involving human subjects were approved by the Institutional Review Boards (IRB) at BGI-Shenzhen. All participants provided written informed consent at enrolment.

**Nasal metagenomic sequencing, quality control**. Metagenomic sequencing was done on the DNBSEQ platform, with 150 bp of paired-end reads for all samples, and four libraries were constructed for each lane. We generated 80.48 ± 23.94 Gb (average ± standard deviation) raw bases per sample for nasal samples[66].

The metapi pipeline (https://github.com/ohmeta/metapi) was used to process the sequencing data. Quality control was first performed with strict standards for filtering and trimming the reads (average Phred quality score ≥ 20 and length ≥ 30) using fastp v0.20.1[87]. After filtering low-quality reads, an average of 77.31 ± 23.00 Gb data was retained with an average of 96.35% host ratio (Supplementary Data 1). Human reads (human genome GRCh38) were then removed using Bowtie2 2.4.2[88].

**Integrating host whole genome sequencing data from blood and nasal samples**. 1,457 individuals with blood samples were sequenced to a mean of 6x for the whole genome. Given that an average of 74.5 Gb (host ratio: 96.35%) of the nasal metagenomic sequencing data were derived from the human genome, we then aggregated the host whole genome sequencing data from blood and nasal samples together for further analysis. The combined reads were aligned to the latest reference human genome GRCh38/hg38 with BWA[89] (v0.7.15) with default parameters. The reads consisting of base quality <5 or containing adaptor sequences were filtered out. The alignments were indexed in the BAM format using Samtools[90] (v0.1.18) and PCR duplicates were marked for downstream filtering using Picardtools (v1.62). The Genome Analysis Toolkit's (GATK[91], v3.8) BaseRecalibrator created recalibration tables to screen known SNPs and INDELs in the BAM files from dbSNP (v150). GATKlite (v2.2.15) was used for subsequent base quality recalibration and removal of read pairs with improperly aligned segments as determined by Stampy. GATK's HaplotypeCaller was used for variant discovery. GVCFs containing SNVs and INDELs from GATK HaplotypeCaller were combined (CombineGVCFs), genotyped (GenotypeGVCFs), variant score recalibrated (VariantRecalibrator), and filtered (ApplyRecalibration). The sensitivity threshold of 99.9% to SNPs and 98% to INDELs were applied for variant selection after optimizing for Transition to Transversion (TiTv) ratios using the GATK ApplyRecalibration command.

We filtered variants to meet these thresholds: (i) Hardy-Weinberg equilibrium (HWE) $p > 10^{-6}$; and (ii) genotype calling rate >98%. We demanded samples to meet these criteria: (i) mean sequencing depth >5×; (ii) variant calling rate >98%; (iii) no population stratification by performing principal components analysis (PCA) analysis implemented in PLINK[92] (v1.9) and (iv) excluding related individuals by calculating pairwise identity by descent (IBD, Pi-hat threshold of 0.1875) in PLINK. After variant and sample quality control, 1,401 individuals with about 7 million common and low-frequency (MAF ≥ 1%) variants were left for M-GWAS analyses.

To evaluate the quality of integrated WGS data, we sequenced 18 blood samples with a high depth of >30X and then assessed concordance rates (CR) of the blood WGS data and integrated WGS data of blood and nasal samples. The average genotype concordance was 98.15% (Supplementary Data 13). Further, PCA analysis showed no population stratification between 1,401 integrated WGS samples in this study and 2002 blood WGS samples as reported previously (Supplementary Fig. 17). PCA analysis also showed all individuals in this cohort (Chinese; nasal_wgs) clustered into the EAS (East Asian) group and obviously separated from the AFR and EUR population from the 1000genome phase 3 dataset (Supplementary Fig. 18).

**Nasal metagenomic taxonomic and functional profiling**. Taxonomy assignment and functional prediction were performed using MetaPhlAn3 version 3.0.7 and HUMAnN3 version v3.0.0.alpha.3 with default settings[93]. The marker gene database used by MetaPhlAn 3 contains ~99,200 fully annotated genomes, including 9,795 bacterial genomes and 122 eukaryotic genomes, which would cover a wide range of microbial diversity, including common nasal cavity fungi such as *Malassezia*. We constructed two MetaPhlAn3 profiles, one using bacteria and fungi together and the other only using bacteria as microbial community, respectively. We evaluated the difference of two profiles. Spearman rank correlation analysis revealed high consistency in the bacteria taxa quantification between the two profiles (mean Spearman correlation rho = 0.9997; Supplementary Fig. 19). Further, we compared the phenotype consistency of bacteria taxa in the M-GWAS analysis, by using profiled only bacteria and using profiled bacteria and fungi together, respectively. This comparison also showed very high consistency between the two different profiled methods, for both abundance-based and presence/absence-based microbial taxa phenotype (Supplementary Fig. 19). All these results suggested that fungi profiled together with bacteria makes little difference for only using bacteria. Thus, we used the MetaPhlAn3 profile including bacteria and fungi together. Finally, we obtained a raw microbial taxonomic dataset composed of a total of 1138 taxa (11 phyla, 27 classes, 53 orders, 103 families, 222 genera, and 722 species) and a functional dataset containing a total of 430 pathways or functions.

**Correlations of host PCs with microbial α-diversity and PCos**. The microbial α-diversity (Shannon and Simpson indices) and β-diversity (Bray–Curtis dissimilarities) were generated based on the species-level abundance data through the 'diversity' and 'vegdist' functions in the R package 'vegan', respectively. Then, principal coordinates analysis (PCoA) was performed based on the calculated beta-diversity dissimilarities using the 'capscale' function in 'vegan'. For each of the top 10 PCs, we used a multivariable linear model to identify its correlation with each α-diversity indices and each PCo with sex, age, BMI and sequencing read counts as covariates. For each of the abundant taxa with abundance over 0.0001, we performed a Wilcoxon rank-sum test to identify the differences between two groups, namely southern and northern Chinese. The assignment of the two groups, southern and northern Chinese, was based on self-reported ancestry information. The questionnaire included the specific question: "Where is the geographical origin of your ancestry (before and including your grandfather)?" The respondents were provided with a list of all provinces of China as possible answers, along with an "unknown" option. We categorized the answered cities into either southern or northern Chinese based on the Qinling and Huaihe River line, which traditionally serve as China's north-south dividing line.

**Association analysis for microbial α-diversity and β-diversity**. GWAS association tests for α-diversity and the top ten PCos were performed using a linear analysis implemented in PLINK v1.9, with sex, age, BMI, sequencing read counts as well as the top ten host genetic principal components (PCs) as covariates. GWAS associations for β-diversity were run using the function manova() from the R 'stats' package, in a multivariate analysis using the same covariates stated above and genotype dosages as derived by PLINK v1.9.

**Association analysis for microbial taxa and functions**. Given the power of GWAS tests, we filter the microbial taxa and functional pathways to keep those with occurrence rates over 10% (present at least 141 individuals) and average relative abundance over $1 \times 10^{-4}$. After filtering, the represented genera of these microbial taxa covered 99.63% of the whole community in the cohort. As many nasal microbial taxa and functions are highly correlated and aim to reduce the number of GWAS tests, we then

performed many Spearman's correlation tests to obtain the independent taxa for M-GWAS analyses. Spearman's correlations were calculated pairwise between all taxa, and the correlations were used to generate an adjacency matrix where correlations of >0.995 represented an edge between taxa. A graphical representation of this matrix was then used for the greedy selection of representative taxa. Nodes (microbiota taxa) were sorted by degree and the one with the highest degree was then chosen as a final taxon (selecting at random in the case of a tie). The taxon and its connected nodes were then removed from the network and the process was repeated until a final set of taxa was found such that each of the discarded taxa was correlated with at least one taxon. These filterings resulted in a final set of 207 microbial features (86 taxa and 207 functions) for association analyses.

We tested the associations between host genetics and oral microbiome using either a linear model based on the relative abundance (AB) or a logistic model based on the presence/absence (P/A) of microbial features. There are 8 taxa and 35 pathways entering into linear models, while the remaining 78 taxa and 172 pathways entered into logistic models (Supplementary Data 4). Specifically, for the 43 microbial features present in over 95% of individuals, their relative abundances were log-transformed that performed better than the centred-log transformed (CLR) data (Supplementary Fig. 20). Then, the residuals were computed using 'lm' with the following covariates in R: (log10(Microbe abundance) _ age + sex + BMI + sequencing read counts + top ten PCs). The residuals from the model were extracted utilizing the function residuals() from the stats package and used in a univariate linear model in the association analysis with genotypes. However, for the 250 microbial feature that appears in >10% but less than 95% of individuals, we dichotomized it into presence/absence patterns to prevent zero inflation, then the abundance of bacteria could be treated as a dichotomous trait for logistic regression analysis with adjusting for the same covariates with above.

We also examined whether other covariates could be an important confounder of results. Except for sex, BMI, and blood metabolites, there are 12 potential host factors statistically associated with nasal microbial beta-diversity ($p < 0.05$, Supplementary Data 10), of which weight is highly correlated with BMI (Spearman rho = 0.45, $p = 8.4e-70$), "Residential area" is highly correlated with PC1(Spearman rho = 0.41, $p = 7.2e-57$). By have been accounting for BMI and PC1 in the GWAS analysis, the effects of weight and residential area are indirectly considered. Then, we did sensitivity analyses by adding each of the 10 other potential confounders for GWAS analysis and observed their effects on the results. After repeating the GWAS for all microbial taxa for which we initially had found at least one genome-wide significantly associated locus, we found that adding any one of the 10 potential cofounders for adjusting had very minor effects on the GWAS association results (Supplementary Data 14). Adding any one of the 10 potential cofounders as covariate did not change the 2 study-wide significant associations and very few of the 180 genome-wide associations changed slightly over the $P = 5 \times 10^{-8}$, which is likely by chance given inclusion of any additional covariate (Supplementary Data 14). In addition, the beta estimates with and without the adjustment of additional 10 covariates were highly consistent (Pearson $r$ ranging from 0.9945 to 0.99999).

We looked through the 2 study-wide significant association signals of nasal GWAS in the gut M-GWAS for comparison. The gut microbiome GWAS data come from our previously published gut M-GWAS paper[24], in which the microbial profile was constructed by aligning sequencing reads to the integrated gene catalogue (IGC) of the gut microbiome. In this paper, for consistency of comparison, we reconstructed the gut microbial

profile using MetaPhlAn3 and did the same M-GWAS analysis as done for the nasal M-GWAS.

We searched the oral microbiome-related SNPs in the summary statistics data from this cohort as previously reported to examine their associations with host traits, as well as to examine their associations with diseases from the Biobank Japan[30,94] database.

**Functional and pathway enrichment analysis of significant signals**. The significant genetic variants identified in the association analysis were mapped to genes using ANNOVAR[95]. Given that some significant genetic variants were low-frequency in this Chinese cohort and not reported by the public 1kgenome database, we thought it was more suitable to input gene lists for enrichment analysis. We mapped variants to genes based on physical distance within a 5 kb window and got the gene lists for enrichment analysis. These genes were calculated expression in a publicly available nasal airway epithelium transcriptome dataset[45] and across the 49 tissues in the GTEx database[46]. The statistical $p$-value < 0.05 was considered statistically significant. In addition, the mapped genes were further investigated using the GENE2-FUNC procedure in FUMA[48] (http://fuma.ctglab.nl/), which provides hypergeometric tests for the list of enriched mapping genes in 53 GTEx tissue-specific gene expression sets, 7,246 MSigDB gene sets, and 2,195 GWAS catalogue gene sets[48]. Using the GENE2FUNC procedure, we examined whether the mapped genes were enriched in specific diseases or traits in the GWAS catalogue, or enriched in specific GO, KEGG, as well as whether showed tissue-specific expression. Significant results were selected if Bonferroni-corrected $p < 0.05$ was observed.

**Environmental factors explained the variance of the oral microbiome**. We next searched for associations between the 340 environmental variables selected above and the oral microbiome compositions. We performed Bray–Curtis distance-based redundancy analysis (dbRDA) to identify variables that are significantly associated with β-diversity and measure the fraction of variance explained by the factors, using the 'capscale' function in the vegan package. The significance of each response variable was confirmed with an analysis of variance (ANOVA) for the dbRDA (anova.cca() function in the vegan package). Only the variables that were significantly associated (Benjamini–Hochberg FDR < 0.05) with the β-diversity estimates in the univariable models were included in the multivariable model. The additive explanatory value (in %) of significant response variables (e.g. environmental parameters, vitamins, and serum amino acids, etc.) was assessed with a variation partitioning analysis of the vegan package ('adj.r.squared' value using RsquareAdj option).

**Host genetics explained the variance of the oral microbiome**. We performed GWAS associations of 7 million variants with β-diversity by using the function manova() from the R 'stats' package and estimated the variance inferred by top 21 lead SNPs using the ordiR2step functions in the vegan package in R. We also performed 100 permutation analyses. In each analysis we (i) randomly assigned to each individual the species profile of a randomly selected individual; (ii) selected lead variants of the top 21 loci according to their association with microbiome β-diversity, using the "manova" function; and (iii) estimated the fraction of β-diversity variance that can be inferred from the top 21 SNPs, using the ordiR2step function. The resulting $p$-value was the fraction of permutations in which the fraction of inferred variance was greater than observed under the real data.

**One-sample MR analysis**. To investigate the causal effects between microbial features and host traits available from this multi-omics cohort, we first performed a one-sample bidirectional MR analysis. We specified a threshold of $p < 1 \times 10^{-6}$ to select SNP instruments and a threshold of LD $r^2 < 0.1$ for clumping analysis to get independent genetic variants as IVs for MR analysis. The use of a looser $p$-value threshold is common in many MR studies, and we additionally calculated F-statistics and variance explained to directly present the strength of our instruments (Supplementary Figs. 13 and 14). Then, an unweighted polygenic risk score (PRS) was calculated for each individual as implemented in PLINK v1.9. Each independent genetic variant was coded as 0, 1, and 2, depending on the number of trait-specific risk-increasing alleles carried by an individual. We performed Instrumental variable (IV) analyses employing the two-stage least square regression (TSLS) method. In the first stage, for each exposure trait, the association between the GRS and observational phenotype value was assessed using the linear regression model and predicted fitted values based on the instrument were obtained. In the second stage, linear regression was performed with outcome traits and genetically predicted exposure levels from the first stage. In both stages, analyses were adjusted for age, sex, sequencing read counts and the top ten principal components of population structure. For each trait, TSLS was performed using the 'ivreg' command from the AER package in R. We next attempted to replicate the causal effects between traits in the replication dataset.

**The growth prediction of bacteria**. To predict the growth of four bacteria, *S. grimesii*, *Y. regensburgei*, *E. bruuniana* and *E. miricola*, we first downloaded the representative genomes from NCBI. The genome-based metabolic models were reconstructed using gapseq[54] version 1.2 with default settings respectively. The combined growth model of these organisms in a shared environment such as cysteine was performed by R package BacArena[96] version 1.8.2.

**Statistics and reproducibility**. Whole metagenomic sequencing was performed for 1,593 nasal samples, 1,457 of which also had blood samples with whole genome sequencing. Taxonomy assignment and functional prediction were performed using MetaPhlAn3 version 3.0.7 and HUMAnN3 version v3.0.0.alpha.3 with default settings. The microbial α-diversity and β-diversity were generated based on the species-level abundance data through the 'diversity' and 'vegdist' functions in the R package 'vegan', respectively. GWAS association tests for α-diversity and the top ten PCos were performed using a linear analysis implemented in PLINK v1.9, with sex, age, BMI, sequencing read counts as well as the top ten PCs as covariates. GWAS associations for β-diversity were run using the function manova() from the R 'stats' package. The associations between host genetics and oral microbiome were tested using either a linear model based on the relative abundance (AB) or a logistic model based on the presence/absence (P/A) of microbial features. Functional and pathway enrichment analysis of significant signals were performed using the "FUMA" tool. We inferred the variance explained by environmental factors and host genetic variants by using the ordiR2step functions in the vegan package in R. The one-sample MR analysis was performed using the 'ivreg' command from the AER package in R. The genome-based metabolic models were reconstructed using gapseq version 1.2 with default settings respectively. The combined growth model of these organisms in a shared environment such as cysteine was performed by R package BacArena version 1.8.2.

**Reporting summary**. Further information on research design is available in the Nature Portfolio Reporting Summary linked to this article.

## Data availability

The data in this study have been deposited to GSA with the project id: PRJCA015657. All GWAS summary statistics data that support the findings of this study including associations between host genetics and nasal microbiome are publicly available in https://ngdc.cncb.ac.cn/gvm/getProjectFile?t=9f187d05 (access id: GVP000013). The nasal metagenomic sequencing data after removing host reads in this study have been deposited to GSA and available in https://ngdc.cncb.ac.cn/gsa-human/browse/HRA004206 (access id: HRA004206). The source data for Fig. 3e, f are in Supplementary Data 15, the source data for Fig. 5a are in Supplementary Data 7, and the source data for Fig. 6 are provided in Supplementary Data 12. The release of these data was approved by the Ministry of Science and Technology of China (Project ID: 2023BAT0694). According to the Human Genetic Resources Administration of China regulation and the institutional review board of BGI-Shenzhen related to protecting individual privacy, the human blood sequencing data are controlled-access and are available via an application on request.

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

## Acknowledgements

We sincerely thank the support provided by China National GeneBank. We thank all the volunteers for their time and for self-collecting the oral samples using our kit. This work was supported by the National Natural Science Foundation of China (No. 32200548).

## Author contributions

T.Z., R.G., and H.J. conceived and organized this study. J.W. initiated the overall health project. X.X., H.Y., Y.Z., W.L., X.J., and L.X. led the organization of the cohort, the sample collection, and the questionnaire collection. M.H. and H.L led the DNA extraction and sequencing. X.L and X.T. processed the whole genome data. L.Z., Y.J., and M.L. processed the metagenome data. X.L., X.T., and L.Z. performed the metagenome-genome-wide association analyses and Mendelian randomization analyses. X.L. wrote the raw manuscript. All authors contributed to data and texts in this manuscript.

## Competing interests

The authors declare no competing interest.
