## [Peer Review File · Communications Biology]

Reviewers' comments:

Reviewer #1 (Remarks to the Author):

Please find my review of the Liu, Tong, Zou, Ju et al. manuscript submitted to Comms Bio and entitled "A genome-wide association study reveals human genetic impact on the nasal microbial diversity, taxa and functions". In this study, authors perform a microbiome-GWAS using bacterial taxa from the nasal cavity of 1401 genotyped Chinese individuals and perform Mendelian Randomization to infer causality between microbial carriage and particular diseases or conditions. The approach is standard nowadays, after a few prominent microbiome GWAS studies have been published, albeit mostly on the gut which makes this study interesting. However, I am communicating some major concerns to be addressed before any consideration for publication.

MAJOR COMMENTS:

Have other covariates than age/gender/BMI been tested and accounted for? I presume there are many other factors that can influence nasal carriage (respiratory infection history, smoking, rural/urban status, asthma/allergy symptoms, sampling time of year, etc?). Gut studies have long suffered (and still are suffering) from incorrect covariate adjustment, which I suppose is something important to establish for nasal and respiratory isolates. You seem to have exhaustive information about this cohort, it would be good to see a covariates analysis and proper adjustment testing (you start to do this in Figure 3e and 3f). On a related note: why highlight geographical differences so prominently (Figure 1) and not account for it as a covariate in the GWAS?

Line 216-218: How well is the fungal classification done? Fungi are typically not lysed the same as bacteria, so care needs to be had if you are to co-extract/purify bacterial and fungal DNA to not have artificially higher or lower levels that are purely due to technical effects and not biology (and yet you seem to have interesting associations with fungal species). Where does this come from and can you provide with an assessment of detected fungal diversity in the nasal niche? Importantly and related to this, you seem to agglomerate together fungal and bacterial diversity (by calling it "microbial") but they are probably affected by different technical effects. Please provide a sanity check of this.

You are using a very exhaustively phenotyped cohort. Could you attempt to explain some of the associations? Recently some gut microbiome GWAS studies have been attempting to do this (Qin et al. 2022 Nat Gen comes to mind). A similar assessment of actual bacterial variation with various traits and phenotypes/extended phenotypes would be very valuable.

Less importance and interpretation should be put in the enrichment analysis results. Those are very conjectural and hypothetic and would need to be tested specifically. Especially as the population structure from these expression repositories can drastically differ from the study population. Please amend with more caution.

MINOR COMMENTS:

The figures are very preliminary and messy, and fail to carry their point across very efficiently. I would have liked to see a little clarification of most of them (specific comments below).

Figure 2 (Manhattan plot): it is quite uninformative to have the list of everything below the plot like that. Probably more interesting would be to annotate the actual plot with interesting results (at the very least, the study-wide significant results). Also, it would be good to have nearest loci name, for a better interpretation and visualisation of results.

Line 179-181 / Figure 3d: Where does this gut microbiome GWAS result come from? Nothing in the methods shows a gut microbiome GWAS being performed.

Figure 4: The figure looks very preliminary and unfinished. Can you provide with clustering on the data to highlight anything worthwhile? Also reduce the amount of text/switch to supplementary

table for details. The figure needs to summarise and I can't see your point here, everything is very messy.

Figure 5: can you highlight the tissues of interest here? Nasal, lung etc?

Regarding the MR analyses: what is the power of this? How good are the genetic instruments? I have a lot of skepticism when I see some traits being tested and compared (e.g. "Marital status", "Residential area", "Nature of work") for which I suspect the genetic instruments are very poor/unsuitable. This makes the whole interpretation of this part of the study very doubtful.

Reviewer #2 (Remarks to the Author):

Review for

"A genome-wide association study reveals human genetic impact on the nasal microbiomal diversity"

In this study, Liu and collaborators explore the influence of host genetics on the nasal microbiome. This is relevant because of the recent attention that the oral and gut microbiomes have gained, with proven influence of the human genetics. But most importantly, because of the potential influence of nasal microbiome in human health, specifically in respiratory and neurodegenerative conditions. Some of the findings go in this direction, although causality is not fully proven. The findings are also interesting as they seem to reflect patterns of population stratification, and the influence of the highlighted genes and associated microbiome on different traits. The methods used are mostly adequate, but the introduction and discussion feel unclear as to what is the aim and value added of the study. The discussion explores the potential implications of the findings but lacks warning and acknowledging of the study limitations. Overall, I found the manuscript interesting and enjoyable. My comments are minor.

Abstract

1. In line 33 "principal components strongly correlated with the nasal microbiota 33 diversity and composition.". specifying which principal components and stating the strength of the correlation will help to convey why is this finding relevant.

2. Line 41-43 "Further observational and Mendelian randomization ...several metabolites such as cystine, cystathionine, and 43 glutamic acid play crucial roles in the host metabolism-nasal microbiota interplays..."

- It is unclear what "observational analyses" point to in this sentence. Were other cohorts included for further analyses with metabolites measurement?

- This sentence feels disconnected from the rest of the abstract and is missing a clear distinction of the directionality of the effect, which is the main objective of Mendelian randomisation (MR) analyses. If the results are not clear in direction and/or not significant enough I suggest removing this claim.

3. Line 43 "This study indicates that the contribution of the host genome to the nasal microbiome is not weaker than that of other host factors." This feels like a more important finding that does not come from the study but rather from a specific analysis. I suggest describing such analysis and highlighting the results more.

Introduction

4. Lines 60-62. While a comprehensive understanding of the nasal microbiome might be lacking (with some studies aiming to this 10.3390/microorganisms11071635), it is not clear how this study will contribute to this issue.

5. Line 69. Remove "An"

6. It might be important to mention in what does this study improve upon these previous studies

7. Line 79. A previous introduction or mention of the 4D-SZ cohort might be needed.

Results

8. The term "self-reported ancestry" is commonly used for ethnical self-identification. Here it seems to refer to the geographical origin of the parents, grandparents? This can be specified here or in the methods section.

9. Line 134 "according to a search" seems to suggest this data is not present in the data-base all the time. If the associations are present in the database just add "as reported in the biobank Japan (BBJ) database"

10. Line 135. A short the description of the possible function of the WNK1 gene would help to explain why it is relevant to mention it.

11. Line 158. The SNP rs138099463 near RND1 appears singled out only because it is associated to a functional pathway and a taxa at the same time. Is there any biological reason that could explain these associations occurring at the same time?. If the pathway is included as covariate in the model of the taxa, does the association with the SNP remain?. If there is no other reason to single this out than a coincidence I suggest leaving this out.

12. A big chunk of the content of the results sections is dedicated to the description of results that are not study-wide significant. While these are important to mention as suggestive results. The authors should make clearer that further investigation is required before including them as conclusions and adding speculation of biological models based on them

13. Line 179. It is very interesting to see that an association found in nasal microbiome was replicated with gut microbiome. This seems to suggest the association is strong enough to not be affected by the difference in microenvironment or tissue. However, the variance explained for Actinomyces by the top 63 SNPs (Supplementary Table 6) does not seem particularly high when compared to the other taxa. Granted, this is a measure dependent on the inner variance of the tissue which could be approached with the beta-diversity of the gut microbiome for comparable measurements. An additional concern is the sudden mention of additional gut samples that were not referenced in any other part of the text. If this is a replication cohort (or sample) it should be properly described or referenced in the methods section

14. Line 188-191. The text tries to convey the idea that the correlation between a phenotype and a microbiome feature supports the genetic associations. This is partly true, but it also brings the risk that other non-genetic factors specific of the cohort could be driving these correlations and creating an artefact to confound the genetic association. Therefore, It is very important to replicate these associations on independent cohorts, or to mention it if not possible and why.

15. Some genetic associations are supported by previous findings in BBJ. Are they also supported in other cohorts UKBB

16. Line 206. Replace "The secondary abundant genus..." with "The second most abundant genus..." and please reference the appropriate table/figure (I believe supplementary figure 6)

17. Compared with the taxa associations, the associated functional pathways have little exploration in their biological mechanisms. The authors could comment on the difficulties to interpret these functional pathways as a single factor pointing to a single biological mechanism.

18. Line 257-259. It is not clear to what pattern the authors are referring. While there is clearly more genes being differentially expressed according to figure 5b, it does not look strikingly different than other tissues. In this sense the term "distinct genetic attribute" feels unearned. I suggest tuning down the conclusion here and stick explicitly to the description of what is observed.

19. Line 283. "...our cohort and that of in FUMA"

20. Lines 313-316. This is one of the most interesting results of this paper and should be highlighted more and if possible, explored more. I suggest adding relative analysis with the inner diversity of other microbiome environments. You could also search for interactions among the 45 significant host factors and the host genetics. A comprehensive interaction model could reveal how these microbial features are shaped.

21. The MR analyses and their interpretations pose too many open possibilities and it is not clear if they all deserve that much attention. First, one-sample MR is prone to bias due to possible unobserved correlation structures specific to the cohort (doi: 10.1093/hmg/ddy163). This could partly explain why the findings show bidirectional effect and not a clear direction of causality. I suggest to add a second sample where possible (for example by taking the SNPs associated to metabolites from a different cohort), and where not possible at all (for example in the case there is no cohort with nasal microbiome information), make note of the limitation and be cautious with the interpretation, until replication is possible. I also suggest being more conservative and keep

only relevant the Bonferroni-corrected associations, while leaving the others out.

Discussion

22. Line 407. Add "larger" to the cohorts invited for the future

23. Line 410-411. This study does not contrast on the previous ones, it rather extends, adding diversity, composition and function.

24. Lines 418-421. The authors tease again about neuropsychiatric diseases, with support of their correlation analyses. Can these disorders (and related traits) included as outcome at any point in the MR analyses?. this should be commented here even if the result is negative.

25. Line 426. I agree that there is a lot of findings considering the sample size and the abundance of the microbial features. This demonstrates the authors made the right choice by using both presence/absence and the abundance, as well as the PCo as variables. Making the most of the little amount of data there is available. My compliments on that.

26. Lines 456-458. This is a striking result that 21 genetic loci can explain nearly the same variance as 45 genetic host factors. It would be interesting to discuss the implications of this in the possible future treatments involving the change the microbiome composition of the nasal cavity. A procedure that has proven rather challenging in the gut microbiome.

27. Line 464. Again The term "observational" is unclear for me in this context.

28. Lines 480-482. This paragraph extends on the results of the MR analysis. The authors should be more conservative and stick to the Bonferroni-corrected results. As such thorough mentions of the effect of these metabolites in other microbiomes should be limited. These last 3 mentions about oral microbiota do not really contribute to the discussion here.

29. I feel something should be mentioned in the discussion about the high correlation PCs and the microbiome diversity. This is an interesting and yet puzzling result because it could be really pointing towards evidence of co-evolution of the microbiome and their host, but it could also be yet another proof of unobserved correlation structures specific to the population. For example, the appropriateness of PCs as markers of population structures has been shown to fail in extreme examples specially with non-European populations, where they are rather a very biased measure of sample structure (<https://www.nature.com/articles/s41598-022-14395-4>).

30. The discussion is missing mentions of the study limitations and interpretation in the light of these. For example: sample size (which could be aided by a statistical power estimation); the limited availability of nasal microbiome data which impedes the replication of most results and severely hinders the interpretation of MR results; the use of PCs and PCos as markers of population structure and microbiome structure; and the integration of two low coverage sequencing data as nasal and blood ADN as a feature, with its respective perks.

Methods

31. Line 517. A host ratio of 96.35% of data was present in the retained data. So only 3.65% is the actual microbiome data

32. Line 544. Please specify how did you use the principal components to measure population stratification. Was it visual inspection?. Comparing with BBJ?. Bounding the first PCs by several standard deviations (3SDs, 5SD)?

33. Line 546. A pi-hat threshold of 0.1875 still leaves relatedness of third degree (~ 0.125) included in the study. A high degree of relatedness can also lead to inflation of the results. I recommend sensitivity analysis with the suggestively significant results using fully unrelated individuals ($\pi\text{-hat} < 0.05$)

34. Line 573. "...two groups, namely southern and northern Chinese". Please specify how where these groups assigned: zip-codes or self-reported?. if self-reported, what was the exact question and what were the possible answers?

35. 606. Being relative abundances, Log-transformation while scales the data, it rarely normalises the data for proper use in regression models. I suggest direct centered-log transformation or inverse-rank transformation after the log transformation. However, I don't want to suggest re-analysis of the full data, but a sensitivity or robustness analysis taking only the most significant results and see if they hold.

36. Line 610. "However, for the microbial feature that appeared in <95% of individuals but 611 over 10% of individuals" is very confusing to read. I suggest "...for the microbial feature that appears in more than 10% but less than 95% of individuals"

Miscellaneous

- The description of the supplementary figures could be more detailed. The difference between supplementary figures 7 and 8 is not clear to me.
- Part of the legends of supplementary tables could be description of the columns, some columns are not clear with the names alone.

Reviewers' comments:

Reviewer #1 (Remarks to the Author):

Please find my review of the Liu, Tong, Zou, Ju et al. manuscript submitted to Comms Bio and entitled “A genome-wide association study reveals human genetic impact on the nasal microbial diversity, taxa and functions” . In this study, authors perform a microbiome-GWAS using bacterial taxa from the nasal cavity of 1401 genotyped Chinese individuals and perform Mendelian Randomization to infer causality between microbial carriage and particular diseases or conditions. The approach is standard nowadays, after a few prominent microbiome GWAS studies have been published, albeit mostly on the gut which makes this study interesting. However, I am communicating some major concerns to be addressed before any consideration for publication.

We thank the reviewer for his/her interest, concise and accurate description of our study, as well as their constructive suggestions throughout the review.

MAJOR COMMENTS:

Have other covariates than age/gender/BMI been tested and accounted for? I presume there are many other factors that can influence nasal carriage (respiratory infection history, smoking, rural/urban status, asthma/allergy symptoms, sampling time of year, etc?). Gut studies have long suffered (and still are suffering) from incorrect covariate adjustment, which I suppose is something important to establish for nasal and respiratory isolates. You seem to have exhaustive information about this cohort, it would be good to see a covariates analysis and proper adjustment testing (you start to do this in Figure 3e and 3f). On a related note: why highlight geographical differences so prominently (Figure 1) and not account for it as a covariate in the GWAS?

We understand the reviewer's concerns about adjusting confounders in the M-GWAS analysis. We included 14 covariates for adjusting, namely age, sex, BMI, sequencing read counts, and top ten host principal components (PCs), as described in the Method (using 'lm' with the following covariates in R: $(\log_{10}(\text{Microbe abundance})) \sim \text{age} + \text{sex} + \text{BMI} + \text{sequencing read counts} + \text{top ten PCs}$). This 4D-SZ cohort is comprised of a

relatively healthy adult population with a mean age of 30. We didn't observe the impact of smoking and sampling time of years on the nasal community. As listed in the Sup table 10, we observed sex and BMI were the most significant factors influencing the composition of nasal microbiome. Not considering the blood metabolites, we found the other factors (lifestyles, diets, etc.) separately showed small effect size associated with the microbiome community (variance explained $R^2 < 0.52\%$ for each factor). Except for sex and BMI, there are 12 potential host factors statistically associated with nasal microbial beta-diversity ($p < 0.05$, Sup table 10), of which weight is highly correlated with BMI (Spearman $\rho = 0.45$, $p = 8.4e-70$), "Residential area" is highly correlated with PC1 (Spearman $\rho = 0.41$, $p = 7.2e-57$). By have been accounting for BMI and PC1 in the GWAS analysis, the effects of weight and residential area are indirectly considered.

Further, we also did a sensitivity analysis by adding each of the 10 other potential confounders for GWAS analysis and observed their effects on the results. After repeating the GWAS for all microbial taxa for which we initially had found at least one genome-wide significantly associated locus, we found that adding any one of the 10 potential cofounders for adjusting had very minor effects on the GWAS association results (new added Supplementary Table 14). Adding any one of the 10 potential cofounders as covariate did not change the 2 study-wide significant associations and very few of the 180 genome-wide associations changed slightly over the $P = 5 \times 10^{-8}$, which is likely by chance given inclusion of any additional covariate (Supplementary Table 14). In addition, the beta estimates with and without the adjustment of additional 10 covariates were highly consistent (Pearson r ranging from 0.9945 to 0.99999). There have been studies suggesting that adjusting for too many covariates decreases power (<https://www.nature.com/articles/ng.2346>). We have included 14 high correlated and necessary confounders and adding too many covariates seem to result in a loss of power. Combined with all these evidences, we thought the current GWAS analysis is reasonable. We have added the Supplementary Table 14 and these details in the Method section.

The reason we didn't include geographical differences as a covariate is that this geographical origin phenotype is highly correlated with PC1 (Wilcoxon rank-sum test, $p < 2.2 \times 10^{-16}$) and PC2 ($p = 1.78 \times 10^{-11}$). By including PC1 and PC2 in the GWAS analysis, it's equivalently that the effect of geographical origin is also accounted for. These principal components are likely capturing some of the underlying genetic and

geographic structure within the dataset.

Line 216-218: How well is the fungal classification done? Fungi are typically not lysed the same as bacteria, so care needs to be had if you are to co-extract/purify bacterial and fungal DNA to not have artificially higher or lower levels that are purely due to technical effects and not biology (and yet you seem to have interesting associations with fungal species). Where does this come from and can you provide with an assessment of detected fungal diversity in the nasal niche? Importantly and related to this, you seem to agglomerate together fungal and bacterial diversity (by calling it “microbial”) but they are probably affected by different technical effects. Please provide a sanity check of this.

This is a good question. We fully agree with the reviewer and understand the difficulty of simultaneously unbiased bacterial and fungal DNA extraction. Studies have reported that bead-beating step increased the recovery of fungal taxa (<https://pubmed.ncbi.nlm.nih.gov/33060634/>; <https://pubmed.ncbi.nlm.nih.gov/32862838/>). In this study, we utilized the MagPure Stool DNA KF Kit B (no. MD5115-02B) for DNA extraction. This protocol includes a step of mechanical cell disruption by bead beating and optimizes the extraction process of DNA from both bacterial and fungal cells while maintaining a higher accuracy compared to other method, as evaluated in our previous study(<https://www.ncbi.nlm.nih.gov/pmc/articles/PMC7355182/>).

After the DNA extraction, we subjected the reads to a filtering process to remove low-quality reads and also removed human reads to focus on the microbial components. We then employed metaPhlAn 3, a gene marker-based taxonomic tool, to assess the relative abundance of fungi and bacteria. The marker gene database used by metaPhlAn 3 contains approximately 99,200 fully annotated genomes, including 9,795 bacterial genomes and 122 eukaryotic genomes, which would cover a wide range of microbial diversity, including common nasal cavity fungi such as *Malassezia*. In this nasal Metaphlan3 profile, we detected an average of 2.976% relative abundance for fungi, which mainly comprised of genus *Malassezia* (2.95% relative abundance). We also evaluated the difference of two profiles constructed using bacteria and fungi together and only using bacteria as microbial community, respectively. Spearman rank correlation analysis revealed high consistency in the bacteria taxa quantification between the two profiles (mean Spearman correlation $\rho=0.9997$; please see the new

added Figure S19). Further, we compared the phenotype consistency of bacteria taxa in the M-GWAS analysis, by using profiled only bacteria and using profiled bacteria and fungi together, respectively. This comparison also showed very high consistency between two different profiled methods, for both abundance-based and presence/absence-based microbial taxa phenotype (new added Figure S19). In a summary, all these results suggested that fungi profiled together with bacteria makes little difference for only using bacteria. We have supplemented these details into the Method section.

In this study, only two fugal species, *Malassezia restricta* and *Malassezia globosa*, entered into the subsequent M-GWAS analysis (supplementary table 4). Only *Malassezia globosa* showed genome-wide genetic association.

Supplementary Fig. 19. The Spearman correlations of two different profiles in bacteria taxa quantification. We constructed two MetaPhlan3 profiles, one using bacteria and fungi together and the other only using bacteria as microbial community, respectively. The comparisons involved in the raw relative abundance of all species, log-transformed relative abundance and Presence/Absence status phenotype used in the M-GWAS analysis.

You are using a very exhaustively phenotyped cohort. Could you attempt to explain some of the associations? Recently some gut microbiome GWAS studies have been attempting to do this (Qin et al. 2022 Nat Gen comes to mind). A similar assessment of actual bacterial variation with various traits and phenotypes/extended phenotypes

would be very valuable.

Yeah. It's indeed exciting and admirable to discover some well interpreted diet-genetics interactions in shaping the microbiota, for example the interaction of LCT-milk diet on *Bifidobacterium* abundance in the gut GWAS study. We have also tried our best efforts to investigate the potential interactions for explaining the 2 study-wide significant associations in this study. As presented in the Main text and shown in the Figure 3, we identified an association between the *CAMK2A* gene and Actinomyces. We also observed that several phenotypes, such as oral ulcers, caries, upper respiratory tract infection and urinary system infection, linked to Actinomyces. Regretfully, there was no evidence for interactions of *CAMK2A* and these phenotypes for the Actinomyces. Similarly, we observed the *POM121L12* locus and several phenotypes (stress sources, frequently allergic sub-health status, gastritis) had separate influences on the *Gemella asaccharolytica*, while we didn't find their interactions on the species. This could possibly be due to the limited sample size of individuals with disease status. This cohort is a relative healthy population with mean age of 30. A younger population is more suitable for studying the genetic effects on the microbiota as they may be less influenced by environmental factors in the aging process.

Less importance and interpretation should be put in the enrichment analysis results. Those are very conjectural and hypothetical and would need to be tested specifically. Especially as the population structure from these expression repositories can drastically differ from the study population. Please amend with more caution.

We appreciate the reviewer's comment and completely agree. It is well-known that functional analysis tools, especially for Asian populations, are lacking. In our study, we utilized the GTEx dataset and GWAS catalog, which are comprehensive and commonly used for functional annotation in GWAS analysis, to investigate the potential implications of the significant loci we identified in diseases and pathways. These datasets were valuable resources in GWAS studies. Furthermore, as a form of replication, we also employed functional annotations from the Japan Biobank. The results demonstrated high consistency between the usage of the GWAS catalog and the Japan Biobank. For instance, the correlations between microbiome-associated variants (MAVs) and asthma, type 2 diabetes (T2D), and colorectal cancer were confirmed through analysis of both the Biobank Japan (BBJ) dataset and the larger GWAS dataset using FUMA. As the reviewer pointed out, we have revised and tuning down the conclusion here and stick explicitly to the observed description.

MINOR COMMENTS:

The figures are very preliminary and messy, and fail to carry their point across very efficiently. I would have liked to see a little clarification of most of them (specific comments below).

We apologized for this. We have revised these Figures as reviewer suggested.

Figure 2 (Manhattan plot): it is quite uninformative to have the list of everything below the plot like that. Probably more interesting would be to annotate the actual plot with interesting results (at the very least, the study-wide significant results). Also, it would be good to have nearest loci name, for a better interpretation and visualisation of results. We have deleted the detailed lists relevant to SNPs and microbial taxa/pathways in the Fig 2. Instead, we visualized results of the two study-wide significant associations with loci names and their associated microbial taxa, as the reviewer suggested.

Line 179-181 / Figure 3d: Where does this gut microbiome GWAS result come from? Nothing in the methods shows a gut microbiome GWAS being performed.

Sorry for that. The gut microbiome GWAS data come from our previously published gut M-GWAS paper (<https://www.nature.com/articles/s41588-021-00968-y>), in which the microbial profile was constructed by aligning sequencing reads to the integrated gene catalog (IGC) of the gut microbiome. In this paper, for consistency of comparison, we reconstructed the gut microbial profile using MetaPhlan3 and did the same M-GWAS analysis as done for the nasal M-GWAS. We then looked through the 2 study-wide significant association signals of nasal GWAS in the gut M-GWAS for comparison. We presumed that some host genetic variants associated with microbes not only in the nasal but also in the gut. We have supplemented these details into the Method section.

Figure 4: The figure looks very preliminary and unfinished. Can you provide with clustering on the data to highlight anything worthwhile? Also reduce the amount of text/switch to supplementary table for details. The figure needs to summarise and I can't see your point here, everything is very messy.

Sorry for that. We have performed a clustering on the data in the Figure 4. As the reviewer suggested, we have simplified the text in the Fig. 4 to make it clearer.

Figure 5: can you highlight the tissues of interest here? Nasal, lung etc?

As the reviewer suggested, we have marked the interested tissue, i.e. nasal airway epithelium, in bold.

Regarding the MR analyses: what is the power of this? How good are the genetic instruments? I have a lot of skepticism when I see some traits being tested and compared (e.g. “Marital status”, “Residential area”, “Nature of work”) for which I suspect the genetic instruments are very poor/unsuitable. This makes the whole interpretation of this part of the study very doubtful.

In the MR analyses, we calculated F-statistics and variance explained to directly present the strength of our instruments (Supplementary Figs. 13 and 14). The mean instrumental F statistics were all greater than 10 (Supplementary Figs. 13, we also showed them here for your convenience), indicating a strong instrumental strength.

Although the IVs of some likely less genetic-associated phenotypes (e.g. “Marital status”, “Residential area”, “Nature of work”) had a power of meeting the conditions of MR Analysis ($F > 10$), we found they have significant lower power compared to metabolic traits (eg. Blood metabolites). Therefore, we fully agree with the reviewer and made some revisions on MR results. We have now only included blood metabolites and some clinical phenotypes (eg. Visible nasal discharge, Urine PH) for MR analysis. These results have been updated in the result section. We hope observational phenotypic correlation analysis and MR results of nasal bacteria and blood metabolites could offer some reference for microbe-target intervention study in the future.

Supplementary Fig. 13. Distribution of the instrumental F statistics for the nasal microbiome and host traits (mainly metabolites) in this study. For each nasal microbiota, we first selected genetic instruments using $p < 1 \times 10^{-6}$. Then, we tested the strengths of these instruments and the mean instrumental F statistics were all greater than 10 (the dotted line), indicating a strong instrumental strength.

Reviewer #2 (Remarks to the Author):

Review for

“A genome-wide association study reveals human genetic impact on the nasal microbiomal diversity”

In this study, Liu and collaborators explore the influence of host genetics on the nasal microbiome. This is relevant because of the recent attention that the oral and gut microbiomes have gained, with proven influence of the human genetics. But most importantly, because of the potential influence of nasal microbiome in human health, specifically in respiratory and neurodegenerative conditions. Some of the findings go in this direction, although causality is not fully proven. The findings are also interesting as they seem to reflect patterns of population stratification, and the influence of the highlighted genes and associated microbiome on different traits. The methods used are mostly adequate, but the introduction and discussion feel unclear as to what is the aim and value added of the study. The discussion explores the potential implications of the findings but lacks warning and acknowledging of the study limitations. Overall, I found the manuscript interesting and enjoyable. My comments are minor.

We appreciate the reviewer’s very positive feedback and recognition of our work, as well as very careful read and constructive suggestions throughout the review. We have revised and improved the introduction to add the value of this study. We have revised the MR part to only keep the 4 *Bonferroni*-corrected significant causal relationships. we have also added the limitations of this study in the Discussion part.

Abstract

1. In line 33 “principal components strongly correlated with the nasal microbiota diversity and composition.” . specifying which principal components and stating the strength of the correlation will help to convey why is this finding relevant.

Thank for pointing out this. As suggested, we have revised the text and specified the top three host genetic principal components strongly correlated with the nasal microbiota diversity and composition.

2. Line 41-43 “ Further observational and Mendelian randomization … several metabolites such as cystine, cystathionine, and glutamic acid play crucial roles in the host metabolism-nasal microbiota interplays…”

- It is unclear what “observational analyses” point to in this sentence. Were other cohorts included for further analyses with metabolites measurement?

We are sorry for the confusion. As we were fortunate to have all the data in the same cohort, we first calculated observational phenotypic correlations of microbes and metabolites and then estimated the directionality of correlation. Observational correlation could be treated as a prerequisite for strong causality. Specifically, our study is more stringent in MR analysis by using two steps. First, we directly calculated phenotypic correlations between microbiome features and mainly metabolic traits, and found 402 observationally significant associations, and limited the causality analyses to these. Second, we used one sample MR to further confirm the direction of 402 observationally significant associations. Based on these two steps, we hope to move from observed associations to causation. Finally, 128 of the 402 observational correlations showed causal effects with suggestive $p < 0.05$, of which 4 were significant after *Bonferroni* correction ($p < 1.24 \times 10^{-4} = 0.05/402$). We have revised the sentence to specify it as “observational correlation” analysis. We have also only reported the *Bonferroni*-corrected MR results as the reviewers suggested.

- This sentence feels disconnected from the rest of the abstract and is missing a clear distinction of the directionality of the effect, which is the main objective of Mendelian randomisation (MR) analyses. If the results are not clear in direction and/or not significant enough I suggest removing this claim.

We apologized for the unclear descriptions. The directionality of the MR effect is very clear. As answered above, we have also only reported the *Bonferroni*-corrected MR results and revised the sentence as “Further observational correlation and Mendelian randomization analyses consistently suggested the influences of *Serratia grimesii* and *Yokenella regensburgei* on cardiometabolic biomarkers (cystine, glutamic acid, and creatine).”

3. Line 43 “This study indicates that the contribution of the host genome to the nasal microbiome is not weaker than that of other host factors.” This feels like a more important finding that does not come from the study but rather from a specific analysis. I suggest describing such analysis and highlighting the results more.

The reviewer is right. We estimated the contributions of host genetics and other factors on the nasal microbiome composition in this work (see Results section “Host genetics and other factors contributed comparably to the nasal microbiome” for details). The 21

top host genetic variants that were most closely associated with β -diversity could explain 10.59% of the variance in the community structure. The 45 significant host factors could infer 10.76% of the variance in the nasal microbiome β -diversity. Thus, we summarized these results as “This study reveals that the contribution of the host genome to the nasal microbiome is not weaker than that of other host factors.” in the Abstract.

Introduction

4. Lines 60-62. While a comprehensive understanding of the nasal microbiome might be lacking (with some studies aiming to this 10.3390/microorganisms11071635), it is not clear how this study will contribute to this issue.

Thank the reviewer’s for very careful reading and very good questions. We have read and cited this paper, and further added the significance of our study in the Introduction section.

5. Line 69. Remove “An”

We have removed it. Thanks.

6. It might be important to mention in what does this study improve upon these previous studies

Thanks for this constructive suggestion. We have reconstructed the introduction, including the limitations of previous studies, supplementing descriptions of this 4D-SZ cohort and what we can do based on this cohort compared with previous study (Lines 60-92).

“These two studies were investigated in individuals with a small sample size. Hence, the contribution of human genes to the composition and functions of the nasal microbiome remains largely unclear. Gaining insight into the factors including genetic and non-genetic that influence and characterize the nasal microbiome within a meticulously designed cohort is essential for comprehending both upper respiratory health and its broader implications for systemic well-being. The 4D-SZ cohort is a carefully designed multi-omics cohort^{19,20,24-27}, comprising shotgun data of the metagenome from multiple body sites including the nasal cavity and host genome. It also incorporates information on metabolic traits, extensive questionnaires, and comprehensive clinical data. Based on the nasal metagenome and host genome data of 1,401 healthy adults from the 4D-SZ cohort, we first estimated the impact of host

genome on the nasal microbial community and demonstrated the significant influence of host genetics principal components on the diversity and composition of the nasal microbiome. Then, we identified genome-wide and study-wide significant associations of host genetic loci with microbial taxa and functional pathways by performing metagenome-genome-wide association study (M-GWAS). We investigated the functional, tissue, and disease enrichments of the nasal microbiome-associated loci. Furthermore, using multi-omics data from the same 4D-SZ cohort, we compared the influence of host genetics and other host factors on shaping the nasal microbiome. Finally, we illustrated the impact of host metabolites such as cysteine on the nasal microbiome through the use of Mendelian randomization (MR). ”

7. Line 79. A previous introduction or mention of the 4D-SZ cohort might be needed. Thanks for this constructive suggestion. As answered above, we had supplemented the introduction of the 4D-SZ cohort.

Results

8. The term “self-reported ancestry” is commonly used for ethnical self-identification. Here it seems to refer to the geographical origin of the parents, grandparents? This can be specified here or in the methods section.

All individuals in this study were Han Chinese. “Self-reported ancestry” here mainly means their ancestry before (including) grandfather were geographically original from southern or northern China. We have specified it in the corresponding place.

9. Line 134 “according to a search” seems to suggest this data is not present in the data-base all the time. If the associations are present in the database just add “as reported in the biobank Japan (BBJ) database”

We have revised it accordingly, thanks.

10. Line 135. A short the description of the possible function of the WNK1 gene would help to explain why it is relevant to mention it.

Thanks for this constructive suggestion. As the reviewer suggested, we added the description of the function of the WNK1 gene as “The serine-threonine kinase WNK1 functioning as a Cl⁻ sensor plays an important role in mature neuron development^{31,32}”.

11. Line 158. The SNP rs138099463 near RND1 appears singled out only because it is

associated to a functional pathway and a taxa at the same time. Is there any biological reason that could explain these associations occurring at the same time?. If the pathway is included as covariate in the model of the taxa, does the association with the SNP remain?. If there is no other reason to single this out than a coincidence I suggest leaving this out.

Yeah, It's a coincidence. As the reviewer suggested, we leaved it out.

12. A big chunk of the content of the results sections is dedicated to the description of results that are not study-wide significant. While these are important to mention as suggestive results. The authors should make clearer that further investigation is required before including them as conclusions and adding speculation of biological models based on them.

We agree with the reviewer that study-wise significant associations would be more desirable ($P < 1.7 \times 10^{-10}$ for 293 microbial features). However, consistent with previous gut M-GWAS from Germany, the Netherlands and Israel (2 Nat Genet, Nature), more abundant signals were detected with genome-wide significance. Because this is the first relatively large M-GWAS for nasal microbiome, we are excited to report also several interesting genome-wide associations, such as genetic SNPs with the two most common nasal genus and SNPs with the fungus. We are clear that these associations need to be further investigated. We have called for further replication of these associations in more and larger cohorts in the discussion. We believe this study offers a good data reference for future host-nasal microbiome interactive studies.

13. Line 179. It is very interesting to see that an association found in nasal microbiome was replicated with gut microbiome. This seems to suggest the association is strong enough to not be affected by the difference in microenvironment or tissue. However, the variance explained for Actinomyces by the top 63 SNPs (Supplementary Table 6) does not seem particularly high when compared to the other taxa. Granted, this is a measure dependent on the inner variance of the tissue which could be approached with the beta-diversity of the gut microbiome for comparable measurements. An additional concern is the sudden mention of additional gut samples that were not referenced in any other part of the text. If this is a replication cohort (or sample) it should be properly described or referenced in the methods section

In this study, we indeed observed that the phenotypes of presence/absence status ($_HB$, logistic model) have lower variance explained than the phenotypes of relative

abundances (_LOGres, linear model). This may be due to statistical difference between linear model and logistic model. We are apologized for the omission about the gut M-GWAS. The gut microbiome GWAS data come from our previously published gut M-GWAS paper (<https://www.nature.com/articles/s41588-021-00968-y>) , in which the microbial profile was constructed by aligning sequencing reads to the integrated gene catalog (IGC) of the gut microbiome. In this paper, for consistency of comparison, we reconstructed the gut microbial profile using MetaPhlan3 and did the same M-GWAS analysis as done for the nasal M-GWAS. We then looked through the 2 study-wide significant association signals of nasal GWAS in the gut M-GWAS for comparison. We presumed that some host genetic variants associated with microbes not only in the nasal but also in the gut. We have supplemented these details into the Method section.

14. Line 188-191. The text tries to convey the idea that the correlation between a phenotype and a microbiome feature supports the genetic associations. This is partly true, but it also brings the risk that other non-genetic factors specific of the cohort could be driving these correlations and creating an artefact to confound the genetic association. Therefore, It is very important to replicate these associations on independent cohorts, or to mention it if not possible and why.

We fully agree with the reviewer that replications of these associations in an independent cohort are very important. But, as we all know, we can't find an available nasal microbiome cohort for replication, especially with host WGS and whole metagenomic data. We have called for further replication of these associations in more and larger cohorts in the discussion section of this paper. We also discussed "the lack of simultaneous nasal microbiome and host whole genome data for replication" as limitations. We expect that as more nasal microbes related studies are conducted; the validation work will become easier.

15. Some genetic associations are supported by previous findings in BBJ. Are they also supported in other cohorts UKBB

Individuals of the 4D-SZ cohort in this study were Chinese. Chinese of this 4D-SZ cohort and Japanese of BBJ cohort were genetically both of Asian ancestry, which was genetically different from the European population, as showed in the PCA plot (new added Figure S18). Therefore, we only replicated associations in BBJ (Biobank Japan) not in UKKB of European ancestry.

16. Line 206. Replace “The secondary abundant genus...” with “The second most abundant genus...” and please reference the appropriate table/figure (I believe supplementary figure 6)

We have revised it accordingly as reviewer pointed out, thanks.

17. Compared with the taxa associations, the associated functional pathways have little exploration in their biological mechanisms. The authors could comment on the difficulties to interpret these functional pathways as a single factor pointing to a single biological mechanism.

We fully agree with reviewer. We have added this in the Main text as the reviewer suggested.

18. Line 257-259. It is not clear to what pattern the authors are referring. While there is clearly more genes being differentially expressed according to figure 5b, it does not look strikingly different than other tissues. In this sense the term “distinct genetic attribute” feels unearned. I suggest turning down the conclusion here and stick explicitly to the description of what is observed.

Thanks for this constructive suggestion. As the reviewer suggested, we have revised this sentence to only describe the observed result.

19. Line 283. “...our cohort and that of in FUMA”

We have revised it as “replicated in this study and that of in GWAS catalog studies using FUMA tool”, thanks.

20. Lines 313-316. This is one of the most interesting results of this paper and should be highlighted more and if possible, explored more. I suggest adding relative analysis with the inner diversity of other microbiome environments. You could also search for interactions among the 45 significant host factors and the host genetics. A comprehensive interaction model could reveal how these microbial features are shaped. We fully agree with the reviewer. It’s indeed exciting and admirable to discover some well interpreted diet-genetics interactions in shaping the microbiota, for example the interaction of LCT-milk diet on *Bifidobacterium* abundance in the gut GWAS study. We have also tried our best efforts to investigate the potential interactions for explaining the 2 study-wide significant associations in this study. As presented in the Main text and shown in the Figure 3, we identified an association between the *CAMK2A* gene and

Actinomyces. We also observed that several phenotypes, such as oral ulcers, caries, upper respiratory tract infection and urinary system infection, linked to Actinomyces. Regretfully, there was no evidence for interactions of *CAMK2A* and these phenotypes for the Actinomyces. Similarly, we observed the *POM121L12* locus and several phenotypes (stress sources, frequently allergic sub-health status, gastritis) had separate influences on the *Gemella asaccharolytica*, while we didn't find their interactions on the species. This could possibly be due to the limited sample size of individuals with disease status. This cohort is a relative healthy population with mean age of 30. A younger population is more suitable for studying the genetic effects on the microbiota as they may be less influenced by environmental factors in the aging process.

21. The MR analyses and their interpretations pose too many open possibilities and it is not clear if they all deserve that much attention. First, one-sample MR is prone to bias due to possible unobserved correlation structures specific to the cohort (doi: 10.1093/hmg/ddy163). This could partly explain why the findings show bidirectional effect and not a clear direction of causality. I suggest to add a second sample where possible (for example by taking the SNPs associated to metabolites from a different cohort), and where not possible at all (for example in the case there is no cohort with nasal microbiome information), make note of the limitation and be cautious with the interpretation, until replication is possible. I also suggest being more conservative and keep only relevant the Bonferroni-corrected associations, while leaving the others out. Although one-sample MR has some bias, it's also with some advantages due to the availability of host genome data, microbiome features, and host phenotypic data in the same cohort. Our study is more stringent by using two steps. First, we performed direct phenotypic correlation analysis between microbiome features and mainly metabolic traits, and found the observationally significant associations, and limited the causality analyses to these. Second, we used one sample MR to further confirm the direct of observationally significant associations. Based on these two steps, we hope to move from observed associations to causations. We are also clear that the MR hits need to be replicated in an independent cohort. But, as we all know, we can't find an available nasal microbiome cohort for replication, especially with host WGS and whole metagenomic data. To address this concern, we discussed these as limitation in the discussion section. We also revised the MR section and focused on only relevant the Bonferroni-corrected MR results, as the reviewer suggested.

Discussion

22. Line 407. Add “larger” to the cohorts invited for the future

We have added it as the reviewer suggested, thanks!

23. Line 410-411. This study does not contrast on the previous ones, it rather extends, adding diversity, composition and function.

We have used the accurate word “extends” instead of “contrast”, thanks.

24. Lines 418-421. The authors tease again about neuropsychiatric diseases, with support of their correlation analyses. Can these disorders (and related traits) included as outcome at any point in the MR analyses?. this should be commented here even if the result is negative.

We previously checked the datasets from the PGC (Psychiatric Genomics Consortium). Only Schizophrenia phenotype is publicly available for Asian population (<https://www.nature.com/articles/s41588-019-0512-x>). We did the MR analysis and were only able to observe a suggestive causal effect of Firmicutes on Schizophrenia (beta=0.145, p=0.25). We have referred to this as the reviewer suggested.

25. Line 426. I agree that there is a lot of findings considering the sample size and the abundance of the microbial features. This demonstrates the authors made the right choice by using both presence/absence and the abundance, as well as the PCo as variables. Making the most of the little amount of data there is available. My compliments on that.

We thank the reviewer for the recognition of our work and all constructive suggestions made on this work.

26. Lines 456-458. This is a striking result that 21 genetic loci can explain nearly the same variance as 45 genetic host factors. It would be interesting to discuss the implications of this in the possible future treatments involving the change the microbiome composition of the nasal cavity. A procedure that has proven rather challenging in the gut microbiome.

We have added some discussion on the potential future implications. By understanding the specific genetic and host factors that influence the nasal microbiome composition,

we can potentially design interventions that target and modify these factors to promote a healthier nasal microbiome. This personalized approach may lead to improved outcomes and more tailored treatments for individuals. Moreover, by considering the interplay between host genetics and the nasal microbiome, we can gain insights into the mechanisms underlying respiratory diseases and their potential treatments. This knowledge may also help identify individuals who are more susceptible to certain conditions, allowing for early interventions or preventive measures.

27. Line 464. Again The term “observational” is unclear for me in this context.

As we were fortunate to have all the data in the same cohort, we first examined the phenotypic correlations between microbes and metabolites and then determined the directionality of these correlations. Observational correlation could be treated as a prerequisite for strong causality. Observational and MR analyses consistently identified 4 significant causal relationships after Bonferroni correction.

28. Lines 480-482. This paragraph extends on the results of the MR analysis. The authors should be more conservative and stick to the Bonferroni-corrected results. As such thorough mentions of the effect of these metabolites in other microbiomes should be limited. These last 3 mentions about oral microbiota do not really contribute to the discussion here.

As answered to the question 21 and as the reviewer suggested, we have revised the MR results and reported only the Bonferroni-corrected significant MR results. We also revised the Main text and only discussed MR results in this nasal microbiome study.

29. I feel something should be mentioned in the discussion about the high correlation PCs and the microbiome diversity. This is an interesting and yet puzzling result because it could be really pointing towards evidence of co-evolution of the microbiome and their host, but it could also be yet another proof of unobserved correlation structures specific to the population. For example, the appropriateness of PCs as markers of population structures has been shown to fail in extreme examples specially with non-European populations, where they are rather a very biased measure of sample structure (<https://www.nature.com/articles/s41598-022-14395-4>).

We agree with the reviewer to highlight the strong correlation between PCs and microbiome diversity. The reviewer raised a good point. This correlation could be really pointing towards evidence of co-evolution of the microbiome and their host, but it could

also be yet another proof of unobserved correlation structures specific to the population. We have added these in the discussion section of our paper. Although the paper referred to by the reviewer indicated possible bias in the PCA analysis, it is important to note that the PCA analysis in this study depicted the Chinese North-South genetic structure (Fig. 1b,c) and showed Chinese significantly separated from the AFR and EUR populations (Supplementary Fig. 18).

30. The discussion is missing mentions of the study limitations and interpretation in the light of these. For example: sample size (which could be aided by a statistical power estimation); the limited availability of nasal microbiome data which impedes the replication of most results and severely hinders the interpretation of MR results; the use of PCs and PCos as markers of population structure and microbiome structure; and the integration of two low coverage sequencing data as nasal and blood DNA as a feature, with its respective perks.

Thanks for this constructive suggestion. We have added these limitations in the Discussion.

“This study has several potential limitations. First, although our study represents the first large nasal M-GWAS conducted thus far, the sample size remains relatively small, similar to early initiated gut M-GWAS studies^{13,15,18}. Unlike host GWAS and MR studies that benefit from cohorts (UKB⁸⁶, BBJ⁸⁷, etc.) with hundreds of thousands of participants, limited sample size of microbiome GWAS reduces statistical power and increases the likelihood of spurious associations. This is an issue in all current M-GWAS studies and could be improved as more microbiome data and host genome increased. Second, in addition to the two study-wide significant associations found in this M-GWAS studies, we have also reported associations that reach the genome-wide significance level. Nevertheless, the lack of simultaneous nasal microbiome and host whole genome data impedes the replication of most GWAS results and severely hinders the interpretation of MR results. Furthermore, it is important to note that the results presented in this study, and similar to most previous gut microbiome studies^{80,88}, lack a clear understanding of the underlying mechanisms through which the microbiome may causally influence the outcomes. Therefore, further replications of results in independent cohorts and explorations of relevant mechanisms are required in the future. Finally, the use of principal components (PCs) and principal coordinate analysis (PCoA) as markers for population structure and microbiome structure may introduce biases. While these approaches offer valuable insights into the genetic and microbial variations among individuals, they may not capture the full complexity of these structures. Additionally, we integrated blood-derived host genome reads and nasal samples-extracted host genome reads to gather the individual’s host genome data. While the average genotype concordance between the direct blood WGS data and integrated WGS data was 98.15%, indicating minimal bias when using the integrated WGS data as a

substitute for blood WGS data, inherent biases may still exist. However, the high genotype concordance and absence of population stratification shown in the PCA analysis demonstrated the feasibility of reconstructing personal genome information from human genome reads in metagenome sequencing data⁸⁹.”

Methods

31. Line 517. A host ratio of 96.35% of data was present in the retained data. So only 3.65% is the actual microbiome data

Yeah, we observed a mean of 96.35% host ratio for the nasal samples, that's why we sequenced a high up to mean 77G data. Such as this, we could ensure a 2.82Gb(=77G*3.65%) data for the nasal microbiome, that is enough for microbiome study (<https://pubmed.ncbi.nlm.nih.gov/35939836/>).

32. Line 544. Please specify how did you use the principal components to measure population stratification. Was it visual inspection?. Comparing with BBJ?. Bounding the first PCs by several standard deviations (3SDs, 5SD)?

We calculated the principal components by integrating this cohort and 1000genomes phase 3 datasets. The PCA plot showed all individuals in this cohort is homogeneous Chinese that clustered into the EAS (East Asian) group and significantly separated from the AFR and EUR population, as expected. We have added Supplementary Figure 18 to show the PCA plot.

33. Line 546. A pi-hat threshold of 0.1875 still leaves relatedness of third degree (~0.125) included in the study. A high degree of relatedness can also lead to inflation of the results. I recommend sensitivity analysis with the suggestively significant results using fully unrelated individuals (pi-hat < 0.05)

The threshold 0.1875 represents the half-way point between 2nd and 3rd degree relatives and is recommended for genetic GWAS by a Nature Protocol paper (<https://www.nature.com/articles/nprot.2010.116>). This threshold is also a common cut-off to use in many papers (<https://ng.neurology.org/content/6/5/e508> , <https://diabetesjournals.org/diabetes/article/62/3/977/15709/Genome-Wide-Association-Study-for-Type-2-Diabetes>) . This cohort is comprised of individuals in a same company participating in annual company physical examination. We have the relationships information of all individuals including their kinships, which were consistent with the genetic analysis result. Therefore, we have excluded all individuals of relatedness. The direct use of pi-hat < 0.05 will cause exclusion of some unrelated individuals and reduce the GWAS power.

34. Line 573. “...two groups, namely southern and northern Chinese” . Please specify how where these groups assigned: zip-codes or self-reported?. if self-reported, what was the exact question and what were the possible answers?

Sorry for the confusion. The assignment of the two groups, southern and northern Chinese, was based on self-reported ancestry information. The questionnaire included the specific question: "Where is the geographical origin of your ancestry (before and including grandfather)?" The respondents were provided with a list of all provinces of China as possible answers, along with an "unknown" option. We categorized the answered cities into either southern or northern Chinese based on the Qinling and Huaihe River line (China's north-south dividing line). We have now included these details in the appropriate section of the Methodology.

35. 606. Being relative abundances, Log-transformation while scales the data, it rarely normalises the data for proper use in regression models. I suggest direct centered-log transformation or inverse-rank transformation after the log transformation. However, I don't want to suggest re-analysis of the full data, but a sensitivity or robustness analysis taking only the most significant results and see if they hold.

Before the GWAS analysis, we compared the normality of log-transformed (green) and centered-log transformed (CLR; red) abundance data for 43 taxa/pathways entering the linear regression analysis. The distribution plots of the Shapiro W-statistic for checking the normality of data showed that the log transformation performed better than the CLR transformation. Therefore, we used direct log-transformation for relative abundances of 43 taxa/pathways. We now have added a Supplementary Fig. 20 and relevant details to the Methods section.

Supplementary Fig. 20. Estimation of the centered-log transformed (CLR) and log transformed W-statistics. Density plot of the Shapiro W-statistics for CLR (red) and log-transformed (green) abundance data for 43 taxa/pathways entering the linear regression analysis. The log-transformed data conformed more strongly to normality with more GWAS tests close to 1.

36. Line 610. “However, for the microbial feature that appeared in <95% of individuals but over 10% of individuals” is very confusing to read. I suggest “...for the microbial feature that appears in more than 10% but less than 95% of individuals”

We have revised it accordingly as the reviewer pointed out, thanks!

Miscellaneous

- The description of the supplementary figures could be more detailed. The difference between supplementary figures 7 and 8 is not clear to me.

Sorry for the unclear descriptions. We have now added the details of supplementary figures for your check. In short, Supplementary Fig. 7 represented the top genes sorted by the number of tissues in which they are significantly expressed. Supplementary Fig. 8 represented the enriched tissues of 413 genes exhibiting suggestively significant nasal MAVs($p < 1e-6$).

- Part of the legends of supplementary tables could be description of the columns, some

columns are not clear with the names alone.

We have supplemented detailed description of the columns in supplementary tables,
thanks!

REVIEWERS' COMMENTS:

Reviewer #1 (Remarks to the Author):

Dear Editor,

I have reviewed the rebuttal from authors of Liu, Tong, Zou, Ju et al. manuscript submitted to Comms Bio and entitled "A genome-wide association study reveals human genetic impact on the nasal microbial diversity, taxa and functions" after a first round of review. The authors addressed all my concerns in quite an exhaustive, clear and detailed manner, with corresponding additions to the manuscript. As far as I can tell, comments from the other reviewer were also very exhaustively addressed.

Congratulations on the authors for an interesting manuscript.

Reviewer #3 (Remarks to the Author):

It is my privilege to read this manuscript as a reviewer. Liu and colleagues performed extensive study to examine the impact of host genetics and factors on nasal microbiome in Chinese populations. This is a well-designed study with rich data and the findings represent an important step toward understanding the factors on nasal microbiome and on host traits. Statistical and bioinformatic methods are well described and largely reasonable. Population stratification is addressed adequately. Unlike other studies, the current study specially reported findings with genome-wide significance and those with study-wide significance given that they tested ~290 microbiome traits, which shall be commended. The manuscript is well written. I would like to congratulate authors for the nice study.

Based on the revised manuscript, I have only one comment on the total contribution of host genetics on beta-diversity for the top 21 variants with $p < 10E-5$ in beta-diversity analysis. This analysis found that these 21 variants collectively explained 10.6% of beta-diversity matrix, a much stronger observation compared to the gut microbiome study published in Nature a few years ago. However, these 21 variants were selected from over 7M variants and the observed 10.5% may be due to the selection procedure. Since the result is highlighted in Abstract, I would encourage the authors to perform permutation analysis (100 times) to estimate how much variation (under null) could be explained by the lead variants of the top 21 loci in each permutation of beta-diversity. For example, if average $R^2 = 2\%$, then the 10.5% is largely informative; however, if average $R^2 = 8\%$ based on permutations, the statement in abstract shall be deemphasized a little bit.